# High-throughput automated methods for classical and operant conditioning of *Drosophila* larvae

Elise C Croteau-Chonka[1,2†], Michael S Clayton[3†], Lalanti Venkatasubramanian[1], Samuel N Harris[3], Benjamin MW Jones[3], Lakshmi Narayan[2], Michael Winding[1,2], Jean-Baptiste Masson[2,4], Marta Zlatic[1,2,3*‡], Kristina T Klein[1,2*‡]

[1]Department of Zoology, University of Cambridge, Cambridge, United Kingdom; [2]Janelia Research Campus, Howard Hughes Medical Institute, Ashburn, United States; [3]MRC Laboratory of Molecular Biology, Cambridge, United Kingdom; [4]Decision and Bayesian Computation, Neuroscience Department & Computational Biology Department, Institut Pasteur, Paris, France

**\*For correspondence:**
mzlatic@mrc-lmb.cam.ac.uk (MZ);
kristina.t.klein@gmail.com (KTK)

[†]These authors contributed equally to this work
[‡]These authors also contributed equally to this work

**Abstract** Learning which stimuli (classical conditioning) or which actions (operant conditioning) predict rewards or punishments can improve chances of survival. However, the circuit mechanisms that underlie distinct types of associative learning are still not fully understood. Automated, high-throughput paradigms for studying different types of associative learning, combined with manipulation of specific neurons in freely behaving animals, can help advance this field. The *Drosophila melanogaster* larva is a tractable model system for studying the circuit basis of behaviour, but many forms of associative learning have not yet been demonstrated in this animal. Here, we developed a high-throughput (i.e. multi-larva) training system that combines real-time behaviour detection of freely moving larvae with targeted opto- and thermogenetic stimulation of tracked animals. Both stimuli are controlled in either open- or closed-loop, and delivered with high temporal and spatial precision. Using this tracker, we show for the first time that *Drosophila* larvae can perform classical conditioning with no overlap between sensory stimuli (i.e. trace conditioning). We also demonstrate that larvae are capable of operant conditioning by inducing a bend direction preference through optogenetic activation of reward-encoding serotonergic neurons. Our results extend the known associative learning capacities of *Drosophila* larvae. Our automated training rig will facilitate the study of many different forms of associative learning and the identification of the neural circuits that underpin them.

## Editor's evaluation

Since classic studies by Pavlov and Skinner about learning and memory in laboratory settings, the field has sought solid ways to use observable behaviors to illuminate how positive and negative reinforcement modulates stimulus-evoked behaviors. This valuable study by Croteau-Chonka et al., represents the latest modernization of such technology, judiciously targeting the highly quantifiable, real-time motor behaviors of the *Drosophila* larva to create a rigorous new paradigm for classical and operant conditioning. The small nervous system of the larva has few parallels for tractability in dissecting various neural mechanisms. What has been lacking, until now, is the technology needed to tackle the neural mechanisms of learning and memory in the maggot with rigorous, high-throughput, and real-time behavioral observations.

## Introduction

Animals must rapidly alter their behaviour in response to environmental changes. An important adaptation strategy is associative learning (*Dickinson, 1981*; *Rescorla, 1988*), in which an animal learns to predict an unconditioned stimulus (US) by the occurrence of a conditioned stimulus (CS). The US is often a punishing or rewarding event such as pain or the discovery of a new food source (*Pavlov, 1927*). The circuit mechanisms that underlie associative learning are still incompletely understood. Furthermore, there are different forms of associative learning and the extent to which distinct circuits underlie distinct types of associative learning is unclear.

The nature of the CS distinguishes two major associative learning types. In classical conditioning (*Pavlov, 1927*), the CS is a stimulus such as an odour. In operant conditioning the CS is the animal's own action (*Skinner, 1938*; *Thorndike, 1911*). Distinct forms of classical conditioning can further be distinguished, for example, based on the timing of CS and US: in delay conditioning they overlap in time, whereas in trace conditioning there is a gap between them (*Dylla et al., 2013*). Systematic identification and comparison of neurons and circuits involved in distinct types of learning paradigms would be greatly facilitated by automated high-throughput training systems combined with optogenetic and thermogenetic manipulation of neurons in freely behaving animals.

We therefore developed an automated training system for classical and operant learning in the tractable genetic model system, the *Drosophila melanogaster* larva. The *Drosophila* larva is particularly well-suited for studying the neural basis of behaviour. Powerful genetic tools have advanced the study of how larval behaviours (*Figure 1A*) are affected by activity at the cellular level. In *Drosophila*, individual neurons are uniquely identifiable, with morphology and function preserved across animals (*Skeath and Thor, 2003*; *Wong et al., 2002*; *Marin et al., 2002*; *Jefferis et al., 2007*). Together with tissue-localised protein expression afforded by binary expression systems like Gal4/UAS and LexA/LexAop (*Fischer et al., 1988*; *Brand and Perrimon, 1993*), this knowledge has yielded neuron-specific drivers (*Jenett et al., 2012*; *Luan et al., 2006*; *Pfeiffer et al., 2010*) that reproducibly target the same groups of cells in each individual. Adding fluorescent markers helps to pinpoint a neuron's location and reveal its anatomical features (*Lee and Luo, 1999*), while expressing light-sensitive channelrhodopsins and temperature-sensitive ion channels enables optogenetic (*Zemelman et al., 2002*; *Lima and Miesenböck, 2005*) or thermogenetic (*Hamada et al., 2008*; *Kitamoto, 2001*) modulation of neuronal activity. Furthermore, the larva's compact central nervous system (CNS) has made it feasible to reconstruct neurons and their synaptic partners from a larval electron microscopy (EM) volume (*Berck et al., 2016*; *Eichler et al., 2017*; *Fushiki et al., 2016*; *Ohyama et al., 2015*; *Schlegel et al., 2016*; *Larderet et al., 2017*; *Jovanic et al., 2016*; *Jovanic et al., 2019*). These reconstructions have given rise to a full wiring diagram of the larval mushroom body (MB) (*Eichler et al., 2017*; *Eschbach et al., 2021*; *Eschbach et al., 2020*) - a region known to play a key role in associative learning (*Aso et al., 2014*; *Honegger et al., 2011*; *Berck et al., 2016*; *Lin et al., 2014*; *Owald and Waddell, 2015*; *Campbell et al., 2013*; *Turner et al., 2008*; *Eichler et al., 2017*).

As with many vertebrates (*Andreatta and Pauli, 2015*; *Brown et al., 1951*; *Jones et al., 2005*; *Braubach et al., 2009*) and invertebrates (*Takeda, 1961*; *Vinauger et al., 2014*; *Alexander et al., 1984*; *Wen et al., 1997*; *Scherer et al., 2003*; *Davis, 2005*; *Cognigni et al., 2018*; *Vogt et al., 2014*), there is overwhelming evidence that *Drosophila* larvae are capable of classical conditioning. They can be trained to approach an odour paired with a gustatory reward (*Schleyer et al., 2011*; *Hendel et al., 2005*; *Kudow et al., 2017*; *Niewalda et al., 2008*), or avoid an odour paired with light (*von Essen et al., 2011*), electric shock (*Aceves-Piña and Quinn, 1979*; *Tully et al., 1994*), heat (*Khurana et al., 2012*), vibration (*Eschbach et al., 2011*), or the bitter compound quinine (*Gerber and Hendel, 2006*; *Apostolopoulou et al., 2014*). A larva's innate avoidance of light and preference for darkness (*Sawin-McCormack et al., 1995*) can also be modulated when paired with reward or punishment (*Gerber et al., 2004*; *von Essen et al., 2011*).

Adult *Drosophila* have also been shown to perform trace conditioning when stimuli do not overlap in time (*Galili et al., 2011*). However, to the best of our knowledge, trace conditioning has never been shown in *Drosophila* larvae, perhaps reflecting genuine learning limitations in this developmental stage. However, the absence of evidence might also reflect the limitations of current methods for studying larval learning. Such methods often deliver a US using optogenetic stimulation, and deliver an olfactory stimulus via natural odour (*Eschbach et al., 2021*). Temporal precision is difficult with these approaches, precluding thorough investigation of larval learning and relative timing of stimuli.

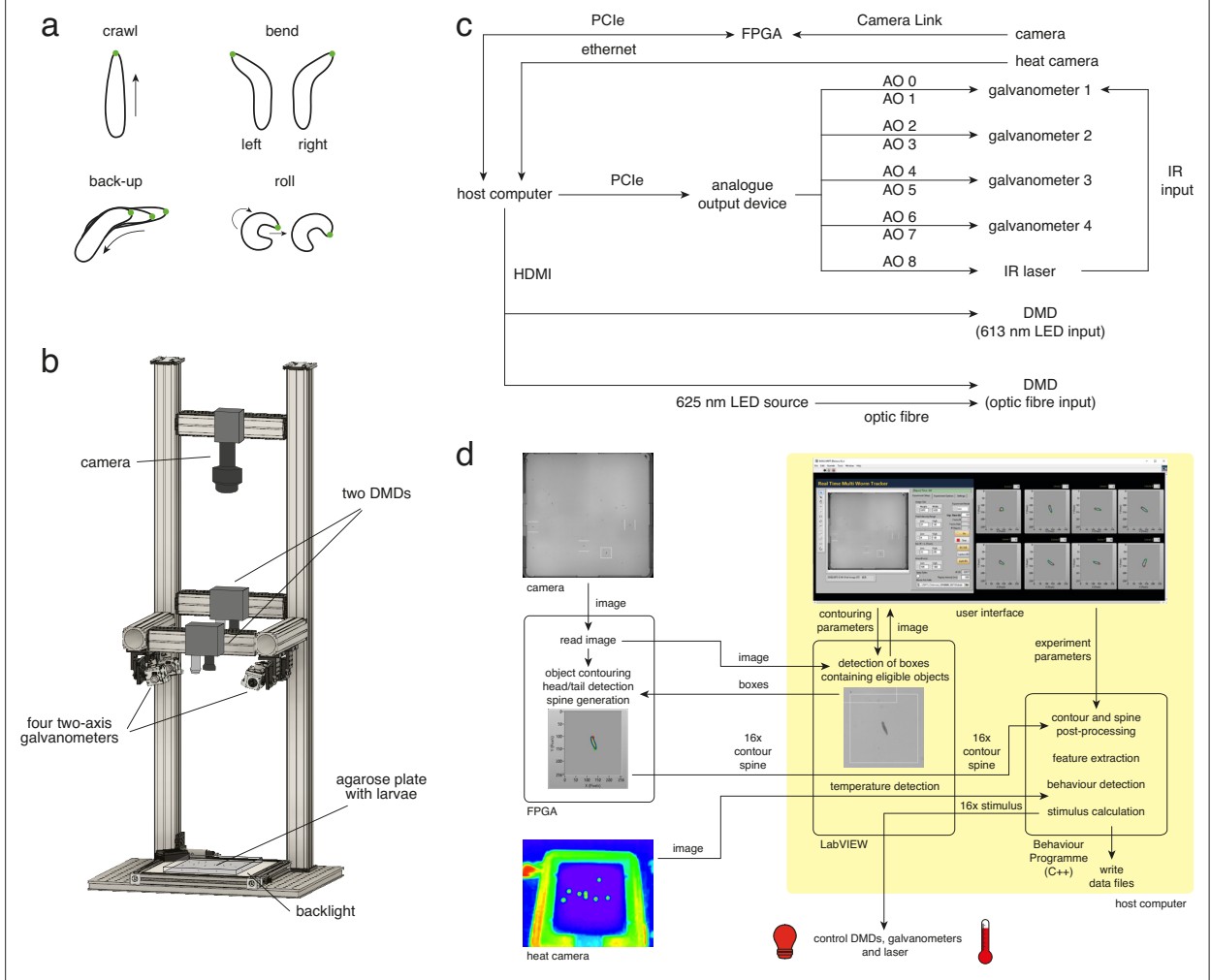

**Figure 1.** Multi-larva tracker combines real-time behaviour detection with either open- or closed-loop stimulation. (**a**) Behavioural repertoire of *Drosophila* larvae. Schematics show the four most prominent actions displayed by *Drosophila* larvae (crawl, left and right bend, back-up, and roll). The larval contour is displayed as a black outline with a green dot marking the head. (**b**) Multi-larva tracker schematic showing the relative positions of the camera, digital micromirror devices (DMDs), galvanometers, agarose plate, and backlight. The heat camera is not shown (for visual simplicity), but is mounted directly beneath the background DMD. See multi-larva-tracker-cad.zip for technical drawings. (**c**) Block diagram of hardware components. AO: analogue output, FPGA: field-programmable gate array. d. Data flow between software elements.

The online version of this article includes the following figure supplement(s) for figure 1:

**Figure supplement 1.** Contour calculation on field-programmable gate array (FPGA).

**Figure supplement 2.** Detecting head and tail.

**Figure supplement 3.** Calculating a smooth spine and landmark points.

**Figure supplement 4.** Calculating direction vectors.

**Figure supplement 5.** Features describing body shape.

**Figure supplement 6.** Velocity features.

**Figure supplement 7.** Temporal smoothing of features.

**Figure supplement 8.** Differentiation by convolution.

In addressing these methodological challenges, prior work has combined optogenetics with thermogenetics to independently stimulate larval neurons of differing sensory modalities (*Honda et al., 2014*). The methods used to heat larvae in these experiments (e.g. hot plates) remain, however, slow relative to optogenetic stimulation. An experimental system equipped with temporally precise

opto- and thermogenetic stimulus delivery would be invaluable for executing automated larval trace conditioning tasks.

Despite countless demonstrations of operant conditioning across vertebrates (*Nottebohm, 1991*; *Olds and Milner, 1954*; *Jin and Costa, 2010*; *Lovell et al., 2015*) and invertebrates (*Brembs, 2003*; *Hoyle, 1979*; *Abramson et al., 2016*; *Nuwal et al., 2012*; *Booker and Quinn, 1981*), it is also unknown whether *Drosophila* larvae can associate their own actions with distinct sensory outcomes. For an animal to learn such a relationship, behavioural information must converge with circuits encoding the outcome's valence. Although vertebrate basal ganglia-like structures exemplify this (*Fee and Goldberg, 2011*; *Redgrave et al., 2011*; *Balleine et al., 2009*), some learned action–outcome associations do not require the brain (*Booker and Quinn, 1981*; *Horridge, 1962*; *Grau et al., 1998*). Investigating operant conditioning in *Drosophila* larvae may further elucidate whether such learning can form in more than one area of the CNS. However, designing an automated operant conditioning task for larvae requires consideration of their short life cycle and physical characteristics. Traditional approaches require either extensive training to interact with an object (*Jin and Costa, 2010*; *Fernando et al., 2015*; *Corbett and Wise, 1980*; *He et al., 2015*) or, as with adult *Drosophila*, partial immobilisation and usage of remaining motion to control a virtual environment (*Nuwal et al., 2012*; *Wolf and Heisenberg, 1991*; *Wolf et al., 1998*; *Brembs, 2011*). A more viable and less restrictive system for larvae would not only allow free animal movement but also rapidly deliver a rewarding or punishing US in accordance with a given behaviour.

Temporally precise action reinforcement requires real-time identification of larval behaviours. Numerous algorithms already exist for real-time tracking of freely moving animals (*Stowers et al., 2017*; *Krynitsky et al., 2020*; *Mischiati et al., 2015*; *Fry et al., 2008*; *Straw et al., 2011*; *Swierczek et al., 2011*) and subsequent offline behaviour analysis (*Mathis et al., 2018*; *Veeraraghavan et al., 2008*; *Dankert et al., 2009*; *Robie et al., 2017*; *Mirat et al., 2013*; *Reddy et al., 2020*; *Stephens et al., 2008*; *Gupta and Gomez-Marin, 2019*). The *Drosophila* larva has been of notable analytic interest due its deformable body and limited set of distinguishing physical features. Algorithmic advances have enabled extensive investigation of larval behaviour in response to open-loop stimuli (*Luo et al., 2010*; *Gershow et al., 2012*; *Denisov et al., 2013*; *Vogelstein et al., 2014*; *Ohyama et al., 2013*; *Ohyama et al., 2015*; *Jovanic et al., 2019*). Most of these approaches are nonetheless not ideal for running in real-time, or require mixing past and future information to perform reliably (*Gomez-Marin et al., 2011*; *Masson et al., 2020*). Existing closed-loop trackers that overcame these challenges to achieve real-time behaviour detection and stimulus presentation are themselves limited to investigating one animal at a time (*Schulze et al., 2015*; *Tadres and Louis, 2020*).

Here, we introduce a new tracker we built to address some methodological limitations of prior *Drosophila* larval learning studies. Our system performs real-time tracking and behaviour analysis of up to 16 larvae simultaneously. It also achieves independent opto- and thermogenetic stimulation through rapid illumination and heating of individual larvae. This stimulation is delivered automatically with high temporal precision and is controlled in either open- or closed-loop. Using this system, we demonstrate that *Drosophila* larvae are capable of both trace conditioning and operant conditioning.

## Results

### Design of an FPGA-based, multi-larva, real-time behaviour-detection and stimulation system for high-throughput, automated training

We built a high-throughput (i.e. multi-larva) tracker combining live behaviour detection with rapid delivery of light and heat stimuli. All hardware resides within an optically opaque enclosure to ensure experiments are performed without environmental light. Larvae move freely on an agarose plate, backlit from below by an infrared (IR) LED and observed from above through a high-resolution camera (*Figure 1B*). While real-time behaviour detection has been developed for a single animal (*Schulze et al., 2015*), our multi-larva system simultaneously tracks up to 16 larvae, using LabVIEW for the user interface and algorithm implementation (*Figure 1D*). Instrumental to this software architecture is the fast image processing speed afforded by field-programmable gate array (FPGA)-based parallelisation (*Soares dos Santos and Ferreira, 2014*; *Li et al., 2011*; *Zhang et al., 2017*). Neuroscientists have adapted FPGA's real-time analysis capabilities (*Kehtarnavaz et al., 2009*; *Uzun et al., 2005*; *Chiuchisan, 2013*; *Yasukawa et al., 2016*) to track rats (*Chen et al., 2005*), zebrafish larvae (*Cong et al.,*

*2017*), and fluorescently labelled neurons in freely behaving *Drosophila* larvae (*Karagyozov et al., 2018*). In our system, the high-performance FPGA and host computer work together (*Figure 1C*) to read raw camera images, detect eligible objects, and extract and process object features (i.e. contour, head and tail position, and body axis) (*Figure 1D*, see Materials and methods for details). Measuring larval body shape, velocity, and direction of motion facilitates robust behaviour detection via machine learning. We detected bends (left and right), rolls, forward and backward peristaltic waves with high precision and recall (see Materials and methods for all values).

The FPGA and host computer also calculate the timing and intensity of light and heat stimuli. Both stimuli can be controlled in either an open-loop or closed-loop configuration. Light stimulation is achieved by directing visible red light through two digital micromirror devices (DMDs), each programmed to project small 1 cm² squares at the location of individual larvae. The DMDs are positioned to project over the entire plate area and operate simultaneously (*Figure 1B*; see also Materials and methods). Targeted heat stimulation of individual larvae can be achieved by directing an 1400–1500 nm IR laser beam through a two-axis scanning galvanometer mirror positioning system, a technique previously used to stimulate single adult flies (*Bath et al., 2014*; *Wu et al., 2014*). The galvanometer's high scanning velocity enables rapid cycling of the IR beam between multiple larvae (*Figure 1B*; see also Materials and methods). To verify the efficiency and speed with which this method heated up individual animals we also installed a heat camera on the setup. Using the camera we demonstrated that larvae are heated to the desired 30°C within 4 s. The heat camera performed closed-loop adjustments of laser intensity to maintain the desired 30°C temperature at each larval location (*Figure 2—figure supplement 1*).

## Proof-of-principle experiments verify multi-larva training rig's stimulation efficiency

We conducted open-loop proof-of-principle experiments to ensure that our tracker could successfully perform optogenetic stimulation of tracked larvae (*Figure 2C*). We tested whether 69F06>CsChrimson and 72F11>CsChrimson larvae rolled upon exposure to red light (*Figure 2C*; see also LABEL:sec:materials_and_methods). Rolling is a lateral movement characterised by the larva curling into a C-shape and quickly turning around its own body axis (*Robertson et al., 2013*; *Hwang et al., 2007*; *Ohyama et al., 2013*; *Figure 1A*). This is the fastest larval escape behaviour and is observed in nature only after exposure to a strong noxious stimulus, such as heat or a predator attack (*Ohyama et al., 2015*; *Robertson et al., 2013*; *Tracey et al., 2003*). Both experimental driver lines drive expression of the red-shifted channelrhodopsin CsChrimson (*Klapoetke et al., 2014*) in neurons whose activation triggers strong rolling behaviour (*Ohyama et al., 2015*): 69F06 drives expression in Goro command neurons for rolling, whereas 72F11 drives expression in the Basin neurons, which integrate mechanosensory and nociceptive stimuli. During the 5 s stimulus presentation within each of three stimulation rounds, we observed above-threshold rolls in over 50% of 69F06>CsChrimson larvae and over 90% of 72F11>CsChrimson larvae. Both experimental groups' rolling behaviour significantly contrasted that of attP2>CsChrimson control larvae during each 5 s stimulus presentation which exhibited virtually no rolling. This significant contrast to the control was also evident for either one or both experimental groups during the first 5 s of break within each stimulation round (*Figure 2D*). These results suggest that the combined red light emitted by the two tracker DMDs (see Materials and methods) is sufficient to activate targeted neurons of interest.

When exploring an environment, a larva alternates between crawling via forward peristalsis (*Heckscher et al., 2012*) and bending its head once or more to the left or right (*Gomez-Marin et al., 2011*; *Luo et al., 2010*; *Kane et al., 2013*; *Figure 1A*). A common avoidance behaviour exhibited by larvae is bending of the head away from undesirable conditions, including extreme temperature (*Luo et al., 2010*; *Lahiri et al., 2011*), light (*Kane et al., 2013*), or wind (*Jovanic et al., 2019*; *Figure 1A*). Prior investigation in our laboratory has shown that Basin activation can evoke bending in addition to rolling, and Goro activation can evoke C-shape bending. We therefore asked whether the fraction of larvae bending during optogenetic stimulation was also significantly different between the experimental groups and control. The fraction of 69F06>CsChrimson and 72F11>CsChrimson larvae that were bending during the 5 s stimulus presentation within each of the three stimulation rounds was significantly higher than that of attP2>CsChrimson control larvae (*Figure 2D*). The low levels of

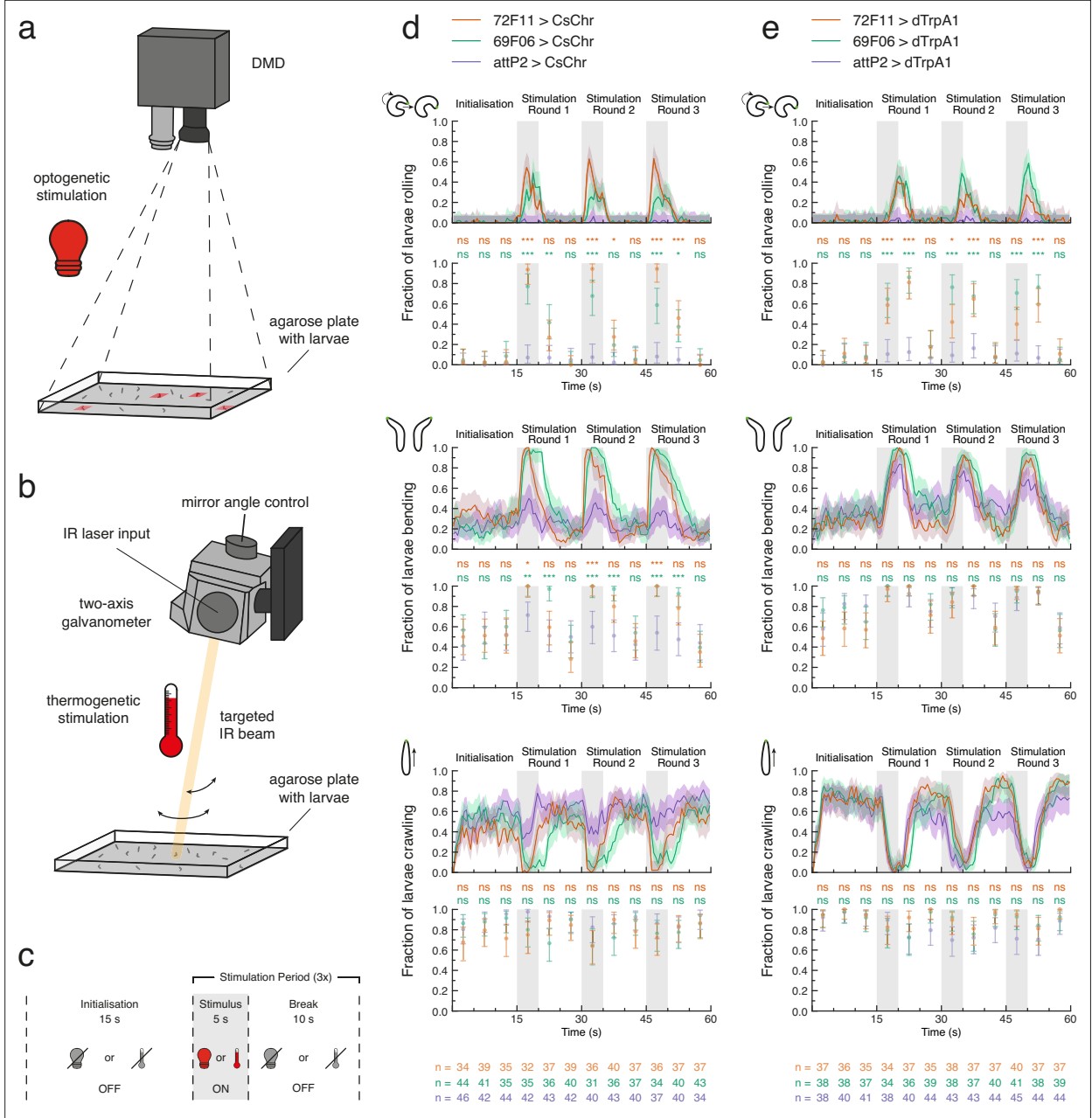

**Figure 2.** Optogenetic and thermogenetic stimulation efficiency verified by behavioural readout. (**a**) Light stimulation hardware schematic. Only one digital micromirror device (DMD) is shown for simplicity. (**b**) Heat stimulation hardware schematic. Only one two-axis galvanometer is shown for simplicity. IR: infrared. (**c**) Proof-of-principle experiment protocol for either optogenetic (light bulb) or thermogenetic (thermometer) stimulation. **d**, **e**. Fraction of larvae for which the optogenetic (**d**) or thermogenetic (**e**) stimulus protocol triggered at least one detected roll (top pair of plots), bend (middle pair of plots), or forward crawl (bottom pair of plots). For each behaviour, the fraction of larvae is computed within 0.5 s (line plots) or 5 s (scatter plots) time bins across the 60 s experiment. All data shown with 95% Clopper-Pearson interval. Fisher's exact test with Bonferroni correction was performed within each 5 s time bin between each experiment group (69F06 and 72F11) and the control group (attP2). Sample sizes for each genotype within each 5 s time bin are shown at the bottom of (**d**) and (**e**). ns p ≥ .05/24 (not significant), * p < .05/24, ** p < .01/24, *** p < .001/24. See *Figure 2— source data 1*, *Figure 2—source data 2*, *Figure 2—source data 3*.

The online version of this article includes the following source data and figure supplement(s) for figure 2:

**Source data 1.** Rolling, bending, and crawling behaviour for each larva over time (separated by genotype) - for data in top row of *Figure 2d, e*.

**Source data 2.** Rolling, bending, and crawling behaviour for each larva over time (separated by genotype) - for data in middle row of *Figure 2d, e*.

**Source data 3.** Rolling, bending, and crawling behaviour for each larva over time (separated by genotype) - for data in bottom row of *Figure 2d, e*.

**Source data 4.** Recorded temperatures during larval IR heating.

*Figure 2 continued on next page*

*Figure 2 continued*

**Figure supplement 1.** Temporal dynamics of larval heating via IR stimulation.

bending evoked by red light alone in control animals suggest the red light used for optogenetic activation is not very aversive to the animals.

The third larval behaviour we monitored across the duration of these optogenetic efficiency experiments was forward crawling. The fraction of larvae crawling was not statistically different between experimental and control groups at any time during the experiment. The fraction of attP2>CsChrimson control larvae crawling during each 5 s stimulus presentation was, however, consistently elevated compared to that of both experimental groups (*Figure 2D*). The reduction in experimental larvae crawling matched our expectations, given the high frequency of both rolling and bending behaviour during stimulus presentation.

We also verified the efficacy of our tracker's galvanometer setup for thermogenetic stimulation (*Figure 2B*). We tested whether 69F06>dTrpA1 and 72F11>dTrpA1 larvae rolled upon exposure to a 1490 nm IR laser (*Figure 2C*; see also Materials and methods). Because wavelengths between 1400 and 1500 nm are well-absorbed by water (*Curcio and Petty, 1951*), we anticipated that heating larvae with this IR beam would activate the ectopically-expressed thermosensitive cation channels (i.e. dTrpA1). During the 5 s stimulus presentation within each of three stimulation rounds, we observed above-threshold rolls in over 60% of 69F06>dTrpA1 larvae and at least 40% of 72F11>dTrpA1 larvae. In stimulation rounds 1 and 2, values in both experiment groups significantly contrasted those of attP2>dTrpA1 control larvae, for which the fraction of larvae rolling was near zero (*Figure 2E*). The quantitative difference in these rolling responses compared to optogenetic activation of the same Gal4 drivers (*Figure 2D*) is not surprising. These effects are likely mediated by differing biophysical properties of CsChrimson and dTrpA1 channels including single channel conductance and open state lifetime (*Pulver et al., 2009*; *Vierock et al., 2017*). In further contrast to the proof-of-principle optogenetic experiments, the slower kinetics of tissue heating caused a ca. 4–5 s second temporal delay between stimulus onset and behaviour onset (*Figure 2E*). We concluded from these results that our chosen heating conditions, although slower than optogenetic stimulation, were effective for targeted dTrpA1 channel activation with a predictable temporal delay of 4 s.

Outfitting our tracker with an IR laser enables targeted neuronal activation without risking the spectral cross-talk commonly associated with channelrhodopsins of overlapping activation wavelengths. Further exploration of the proof-of-principle thermogenetic data did, however, reveal an important caveat of our IR-induced stimulation approach. The fraction of attP2>dTrpA1 control larvae that were bending was statistically indistinguishable from that of 69F06 and 72F11 larvae throughout the experiment (*Figure 2E*). Bending together with rolling accounted for much of larval behaviour across all three groups during each 5 s stimulus presentation; evidenced, in part, by the near-zero fraction of larvae crawling at these times (*Figure 2E*). The high bending frequency of attP2>dTrpA1 control larvae during IR-induced stimulation is likely indicative of mild heat aversion.

## Aversion to fictive Or42b develops after forward-paired trace conditioning

Having verified the efficacy of optogenetic and thermogenetic stimulation in our system, we first studied whether these methods could be used to train larvae in a previously unexplored classical conditioning task that requires precise temporal control of both CS and US. In particular, we focused on trace conditioning, which has not been demonstrated previously in larvae. To provide the CS in these experiments, all larvae expressed CsChrimson in Or42b neurons. *Drosophila* larvae display innate attraction during Or42b activation. When these neurons are stimulated artificially, larvae reduce their bending frequency immediately after stimulation onset and rapidly increase bending frequency following stimulation offset (*Gepner et al., 2015*). To provide the US in these experiments, experimental larvae also expressed dTrpA1 in Basin neurons. Pairing an odour with Basin activation has previously been shown to evoke aversive odour memory (when the CS and US overlap in time *Eschbach et al., 2020*). Importantly, control larvae did not express dTrpA1 in Basin neurons (see Materials and methods for more details).

*Figure 3A* schematises the classical conditioning protocol. Each forward-paired training round comprised 20 s of optogenetic Or42b activation followed by 20 s of IR heating (activating Basin

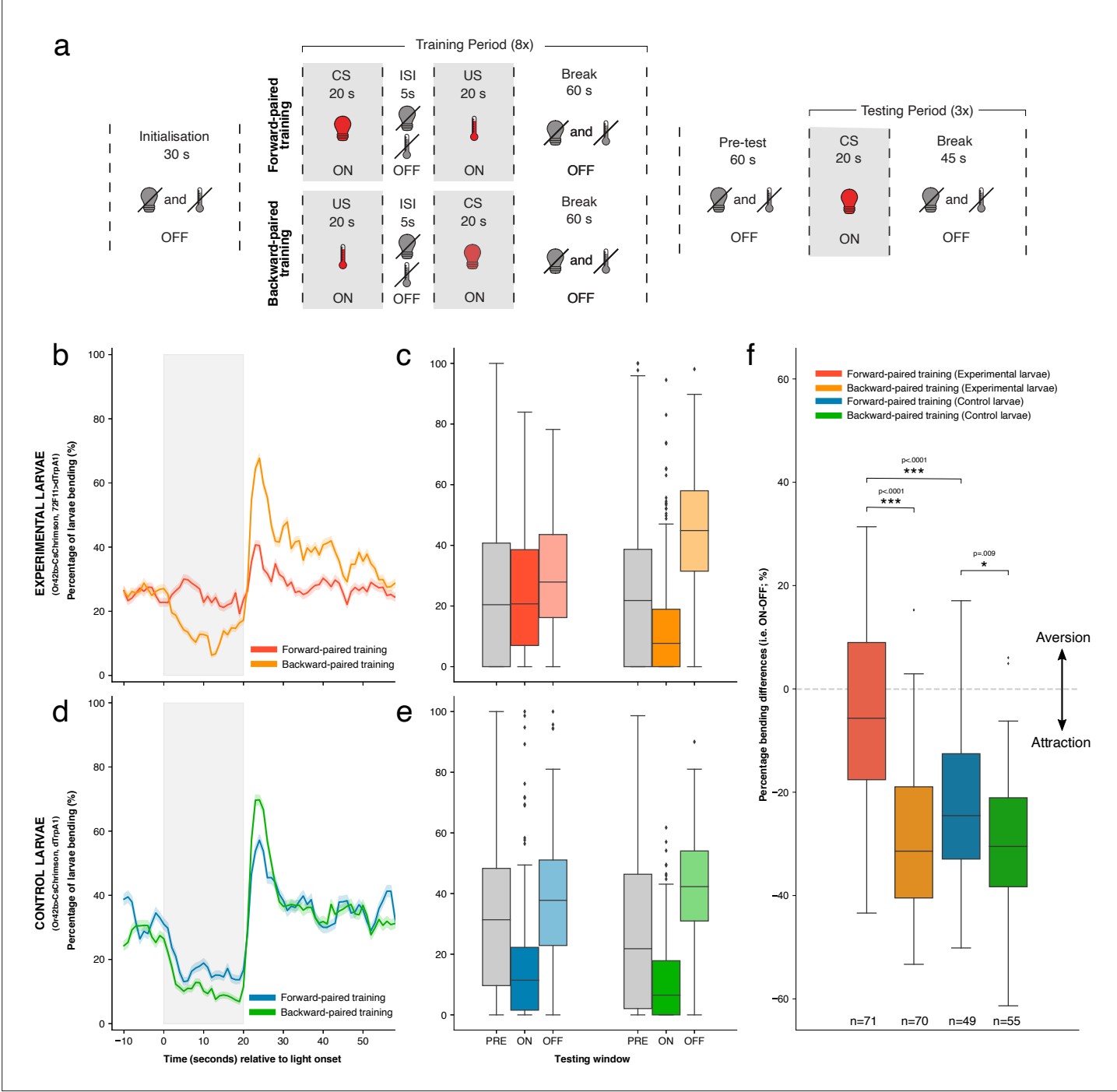

**Figure 3.** The effects of forward- versus backward-paired aversive training on larval attraction to Or42b. (**a**) Schematic of classical conditioning protocol. After an initialisation period of 30 s, the first training round began. Here, Or42b was activated through red light illumination and was followed (forward-paired) or preceded (backward-paired) by the activation of Basin (72F11) neurons through heating. These stimuli were each delivered for 20 s, with a 5 s gap between them (i.e. inter-stimulus interval; ISI). A break of 60 s was allowed before the start of the next training round. In total, larvae completed eight training rounds (i.e. one training period). Larvae then completed a 60 s pre-test period without stimulation before the start of the testing period. The testing period comprised three testing rounds. During a single testing round, only Or42b was activated through red light illumination for 20 s, followed by a 45 s break. (**b**) Time-course of the percentage of experimental larvae bending during the testing period, averaged across all three testing rounds. Data were down-sampled from 20 Hz to 1 Hz to aid visualisation. Grey shading indicates the period of Or42b stimulation. Error shading shows the mean ± 95% confidence intervals (**c**) The average percentage of experimental larvae bending during the PRE (10–0 s before light onset), ON (0–20 s after light onset), and OFF (0–20 s after light offset) testing windows. Bars show medians, as well as upper and lower quartiles of data. (**d, e**) Data presented as in b and c, but for control larvae. (**f**) Percentage bending difference (ON-OFF) values after forward- and backward-paired training for both

*Figure 3 continued on next page*

*Figure 3 continued*

experimental and control larvae. Bars show medians, as well as upper and lower quartiles of data. Statistics calculated with a two-sided Mann-Whitney *U* test; * p < .01, *** p < .0001. See **Figure 3—source data 1**.

The online version of this article includes the following source data for figure 3:

**Source data 1.** Behavioural data recorded during the testing period of associative conditioning experiments.

neurons in experimental larvae), with 5s from light offset to triggering IR stimulation. We note that, while we triggered IR light 5 s after the offset of red light for optogenetic stimulation, it took a further 4 s for larvae to reach the appropriate 30°C temperature for thermogenetic activation of Basins. The gap between CS offset and US onset is, therefore, ca. 9 s. 60 s without stimulation followed US offset. Backward-paired training followed the same protocol structure except IR heating preceded Or42b activation. Eight replicate training rounds comprised the training period, after which an additional 60 s without stimulation was allocated before the testing period. The testing period included three, 20 s blocks of Or42b activation, each separated by 45 s without stimulation (see Materials and methods for more details). To assess learning, we analysed the behavioural responses of larvae to Or42b activation in the testing period. We calculated the difference in the percentage of larvae bending during versus after Or42b activation (i. e. ON-OFF; see Materials and methods for more details). A large, negative 'percentage bending difference' indicates significant attraction to Or42b activation, with less bending during versus after stimulation. In contrast, a difference value closer to zero indicates less attraction and a value greater than zero indicates aversion.

Experimental larvae showed attraction to Or42b activation after backward-paired training, bending less to fictive odour onset and more to odour offset in the testing period. However, after forward-paired training, experimental larvae showed reduced attraction to Or42b (*Figure 3B*). This effect can also be seen in the averaged percentage of larvae bending for each of the three testing windows (*Figure 3C*). We statistically confirmed this qualitative difference between training conditions for experimental larvae. Percentage bending differences were significantly smaller after forward-paired training (mean = −5.22%, sd = 18.23%, n = 71) compared to backward-paired training (mean = −29.65%, sd = 14.10%, n = 70) (p<0.0001, common language effect size (CLES)=0.852; *Figure 3F*). This result matched our expectation, given existing knowledge that Basin activation alone produces an aversive response and that larvae avoid odours paired with Basin activation (*Eschbach et al., 2020*).

The responses of control larvae to Or42b activation were more similar between training conditions (*Figure 3D*, *Figure 3E*). However, as with experimental larvae, percentage bending differences were also significantly smaller after forward-paired training (mean = −21.76%, sd = 15.63%, n = 49) compared to backward-paired training (mean = −30.22%, sd = 14.83%, n = 55), although with a substantially reduced effect size (p=0.009, CLES = 0.648; *Figure 3F*). This difference between training conditions in control larvae suggests that the delivery of mild heat after Or42b stimulation was sufficient to weaken larval attraction to this fictive odour. This result is consistent with prior work showing larval classical conditioning using mild heat as reinforcement (*Khurana et al., 2012*). Importantly, however, forward-paired training yielded significantly smaller percentage bending differences in experimental larvae compared to control larvae (i. e. mean = −5.22% vs −21.76%) (p<0.0001, CLES = 0.752; *Figure 3F*). This result shows that activating Basin neurons after Or42b caused significantly greater aversion to this fictive odour, as opposed to delivering mild heat alone. Bending differences following backward-paired training did not differ significantly between genotypes (p>0.5).

These results show that our tracker can be used to perform automated, high-throughput classical conditioning in *Drosophila* larvae. To the best of our knowledge, these results also provide the first evidence that larvae can perform classical conditioning with significant offset-to-onset gaps (9 s) between stimuli (i.e. trace conditioning).

## Operant conditioning of larval bend direction

Our system's closed-loop stimulation capabilities allowed us to investigate whether *Drosophila* larvae are capable of operant learning. We chose fictive activation of candidate reward circuits as a US (*Figure 4A*), but were challenged to determine which neurons could convey a reinforcement signal for operant learning. Across the animal kingdom, biogenic amine neurotransmitters including dopamine, octopamine, and serotonin can provide reinforcement signals (*Giurfa, 2006*; *Hawkins and Byrne,*

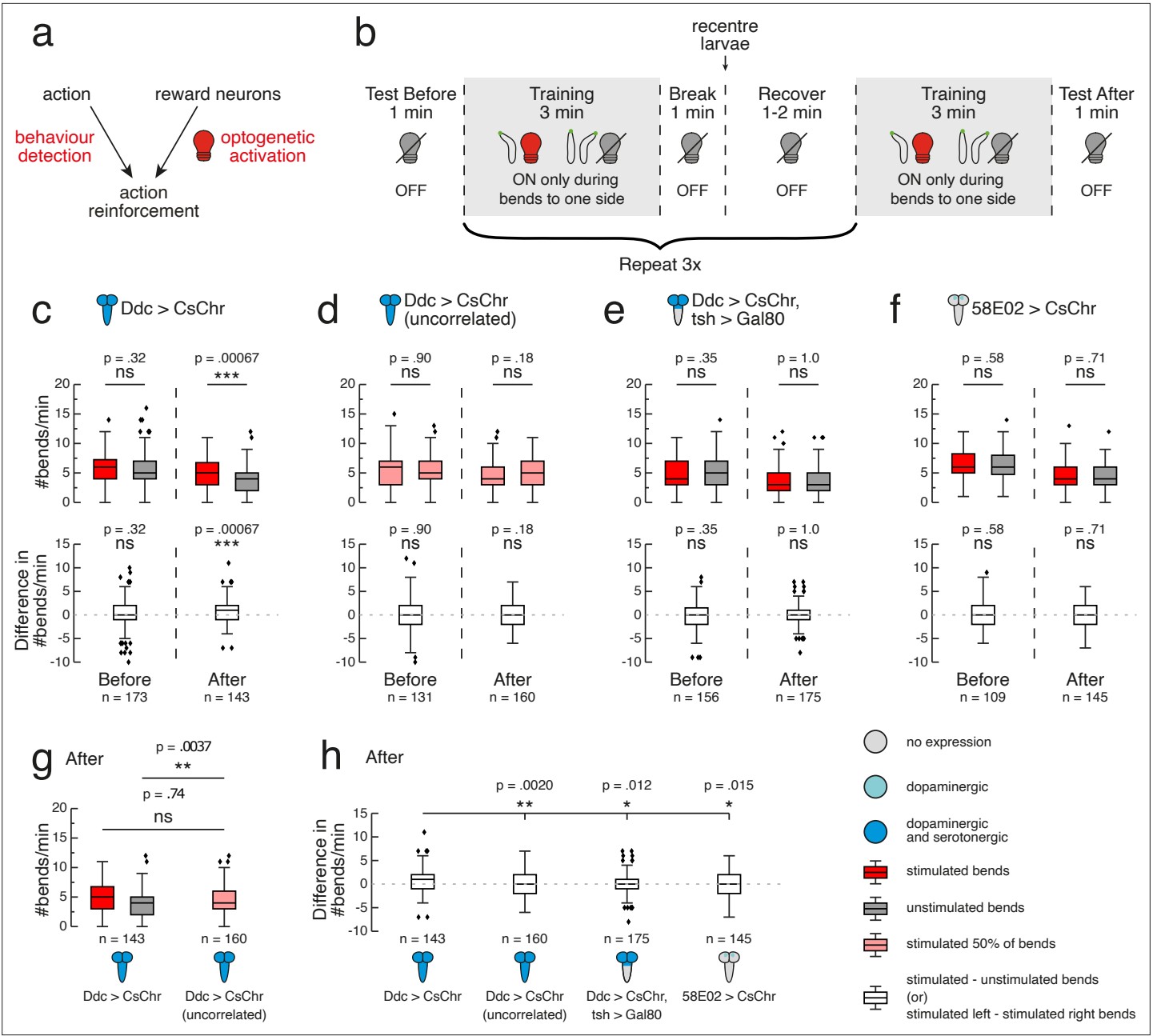

**Figure 4.** Operant conditioning of bend direction in *Drosophila* larvae requires the ventral nerve cord. (**a**) The goal of our automated operant conditioning paradigm is to reinforce an action of interest by coupling real-time behaviour detection with optogenetic activation of reward circuits. (**b**) High-throughput experiment protocol. During training, each larva (black contour with green head) received optogenetic stimulus (red light bulb) when bent to one predefined side (depicted as left), and no stimulus otherwise (grey light bulb). (**c–h**) Gal4 expression is depicted as color-coded CNS (see legend). UAS-CsChrimson effector abbreviated as CsChr for visual clarity. Bars show medians, as well as upper and lower quartiles of data. Outliers (filled diamonds) are randomly jittered horizontally to aid visualisation. (**c–f**) top row. Larval bend rate shown as number of bends per minute, grouped by bends to stimulated side (dark red) or unstimulated side (grey). For larvae that received random, uncorrelated stimulation during 50% of bends (**d**), left and right bend rate are shown in light red. Statistical comparisons calculated using a paired, two-sided Wilcoxon signed-rank test. (**c–f**) bottom row. Difference in bend rate (black) shown between the stimulated and unstimulated sides or, in the case of the uncorrelated training group (panel **d**), between left and right sides. Statistical comparisons calculated using a two-sided Wilcoxon signed-rank test. (**c–f**). Data shown from the test periods before training round 1 (Before) and after training round 4 (After). n is the number of larvae in each time bin. Exact p-values written above corresponding data. ns p ≥ .05 (not significant), * p < .05, ** p < .01, *** p < .001. (**g**). Bend rate data after training round 4 (same data as in (**c**) and (**d**) top row), with bend rate for uncorrelated training group calculated without stratification by bend direction. Statistics calculated with a two-sided Mann-Whitney *U* test, with Bonferroni correction; ns p ≥ .05/2 (not significant), ** p < .01/2. (**e**) Difference in bend rate after training round 4 (same data as in **c–f** bottom row).

*Figure 4 continued on next page*

*Figure 4 continued*

Statistical comparisons against Ddc > CsChr calculated with a two-sided Mann-Whitney *U* test, with Bonferroni correction; * p < .05/3, ** p < .01/3. See *Figure 4—source data 1*.

The online version of this article includes the following source data and figure supplement(s) for figure 4:

**Source data 1.** Source data showing that operant conditioning of bend direction in *Drosophila* larvae requires the ventral nerve cord.

**Source data 2.** Source data showing that *Drosophila* larvae exhibit bend direction preference during operant paradigm training.

**Figure supplement 1.** Ddc-Gal4 expression pattern without and with tsh-Gal80 restriction.

**Figure supplement 2.** *Drosophila* larvae show bend direction preference during operant paradigm training.

**Figure supplement 3.** Operant conditioning of bend direction in *Drosophila* larvae with single-larva tracker.

*2015*; *Meneses and Liy-Salmeron, 2012*; *Fee and Goldberg, 2011*). With this knowledge, we aimed to induce larval operant conditioning by stimulating both dopaminergic and serotonergic neurons across the *Drosophila* CNS. The Ddc-Gal4 driver is ideally suited for this purpose, covering a broad set of neurons containing these biogenic amines (*Li et al., 2000*; *Sitaraman et al., 2008*; *Lundell and Hirsh, 1994*), including the MB-innervating protocerebral anterior medial (PAM) cluster dopaminergic neurons (*Liu et al., 2012*; *Aso et al., 2012*). Although the function of most Ddc neurons is unknown, PAM cluster activation serves as both a necessary and sufficient reward signal in classical conditioning (*Rohwedder et al., 2016*; *Liu et al., 2012*; *Vogt et al., 2014*; *Cognigni et al., 2018*; *Waddell, 2013*), and collective activation of Ddc neurons can substitute for an olfactory conditioning reward in adult flies (*Liu et al., 2012*; *Shyu et al., 2017*; *Aso et al., 2012*). If either dopamine or serotonin mediates valence signalling in larval operant conditioning, paired activation of Ddc neurons with behaviour may be sufficient to induce such learning.

The CS in our automated operant conditioning paradigm was larval bending (*Figure 4B*). We expressed UAS-CsChrimson under the control of the Ddc-Gal4 driver, with the intention of activating corresponding neurons via optogenetic stimulation during bends to a predefined side. Our goal was to establish a learned direction preference, conditioning Ddc>CsChrimson larvae to bend more often to one side than the other. The methodological choice of optogenetics was informed, in part, by a deeper investigation of larval heating dynamics as they relate to average bend duration. We determined that our IR stimulation hardware takes approximately 4 s to heat larval tissue to the nearly 30°C required for dTrpA1 channel activation (*Figure 2—figure supplement 1*; see also LABEL:sec:materials_and_methods). Knowing that the average duration of a larval bend is only a third of that time (mean = 1.35 s, sd = 1.67 s, n = 4622 bends), we concluded that closed-loop heat stimulation would activate neurons of interest only after a noticeable delay relative to behaviour detection, with possibly other behaviours occurring during the delay period. Such a task in which different behaviours are occurring prior to reinforcement could be very difficult to learn. Quicker heating was achievable with increased laser intensity, but such an approach risked overshooting the desired temperature and damaging larval tissue (data not shown). Any safe thermogenetic approach would therefore be too slow to temporally align the US induced via larval heating with a specific larval action. A second important methodological consideration was the risk of establishing conflicting valence signals. We wanted to avoid mixing punishment (via IR-induced tissue heating) with reward (via IR-induced activation of Ddc neurons) during training. With mild heat more aversive to larvae than visible red light (compare bending in control larvae in *Figure 2D, E*, ), we favoured optogenetics over thermogenetics for our operant conditioning paradigm.

Although the direction of bending that triggered optogenetic stimulation was randomised across operant conditioning trials, we summarise the experiment procedure for which this predefined side was the larva's left (*Figure 4B*). Each experiment began with a 1 min test period without red light presentation. Four training rounds followed (each 3 min long) in which larvae received optogenetic stimulation for the full duration of every left bend. The time between the tracker detecting a left bend and light onset was no longer than 50 ms. Between training rounds, larvae experienced 3 min without stimulation. This time was used to brush larvae back to the centre of the agarose plate. We performed this recentring to mitigate the experimental side effects of larvae reaching the plate's edge (see Materials and methods for more details). Following the fourth training round was a 1 min test period without stimulation (*Figure 4B*). For each larva, bend rate, measured as the number of bends per minute performed towards a given side, served as a read-out for bend direction preference. The

difference in bend rate between the stimulated and unstimulated side was also calculated for each larva. This equates to the number of bends per minute to the stimulated side minus the number of bends per minute to the unstimulated side. Within a given time bin (see Materials and methods for more details), the statistical test comparing the difference in bend rate to zero is mathematically equivalent to comparing bend rates against one another. However, computing the difference in bend rate facilitated comparison between genotypes that themselves may differ in basal bend rate.

In the 1 min test before the first training round, we observed no significant difference in bend rate to either side for Ddc>CsChrimson larvae (p>0.5; *Figure 4C*). Larval bend rate to the stimulated side was significantly greater than that of the unstimulated side throughout the majority of training, with this difference tending to widen over the course of each round (*Figure 4—figure supplement 2*). These findings suggest that Ddc activation can function as a rewarding stimulus that larvae increasingly seek with time. In the 1 min test after the fourth training round, larvae showed a significantly greater bend rate towards the side paired with red light stimulation during training (mean = 4.84, sd = 2.40, n = 143) compared to the previously unstimulated side (mean = 4.10, sd = 2.23, n = 143) (p=0.0007, CLES = 0.59; *Figure 4C*) indicating operant conditioning of bend direction.

We also used a previously developed, low-throughput, single-larva, closed-loop tracking system to test the reproducibility of this result on a different system (see Materials and methods for more details) (*Schulze et al., 2015*). Fictive Ddc activation with this system also yielded a significant bend direction preference to the previously stimulated (mean = 5.77, sd = 2.71, n = 109) versus previously unstimulated (mean = 4.73, sd = 2.73, n = 109) side (p=0.0043, CLES = 0.63), after training. These results contrast those of effector control larvae that had the UAS-CsChrimson transgene but not the Ddc-GAL4. The control larvae show no difference in bend rate to either side after training. Based on these control larvae, we concluded that potential leaky expression of CsChrimson in neurons outside of the Ddc expression pattern is not causing operant learning (*Figure 4—figure supplement 3*).

To confirm that the observed bend direction preference was attributable to pairing Ddc activation with bends solely in one direction, we conducted another control experiment in our high-throughput multi-larva training rig, in which larvae received random, uncorrelated stimulation during 50% of all bends. Before, during and after training, these larvae showed no difference in absolute left and right bend rates (p>0.5), with the exception of the first minute of training round 4 (*Figure 4D*, *Figure 4—figure supplement 2*). These larvae also showed a significantly lower difference between left and right bend rates after training (mean = -0.28, sd = 2.73, n = 160) compared to pair-trained larvae (mean = 0.74, sd = 2.62, n = 143) (p=0.0020, CLES = 0.60; *Figure 4H*). Further dissection of bend rates to each side showed that, after training, bend rates averaged together for larvae that received uncorrelated training (mean = 4.80, sd = 2.39, n = 160) were indistinguishable from the rate of pair-trained larvae bending to the previously stimulated side (mean = 4.84, sd = 2.40, n = 143) (p>0.5; *Figure 4G*). However, larvae that received uncorrelated training showed a significantly higher bend rate (mean = 4.80, sd = 2.39, n = 160) compared to pair-trained larvae bending to the previously unstimulated side (mean = 4.10, sd = 2.23, n = 143) (p=0.0037, CLES = 0.58; *Figure 4G*). This suggests that the pair-trained Ddc>CsChrimson larvae have learnt to avoid the unstimulated side. There is growing evidence from classical conditioning that larvae can learn that an unpaired stimulus predicts the absence of reinforcement (*Schleyer et al., 2018*; *Eschbach et al., 2020*). Perhaps larvae are also forming memories of opposite valence in our operant conditioning paradigm, bending less to the unstimulated side because bending to that side predicts the absence of appetitive Ddc activation.

## The mushroom body is not sufficient to mediate operant conditioning in larvae

Our experiments showed that fictive activation of Ddc neurons is a sufficient US for operant conditioning. While we did not identify which individual neurons mediate the observed effect, we hypothesised that not all Ddc neurons are involved. Some prior work in adult flies suggests that the MB is involved in operant conditioning (*Sun et al., 2020*), while other studies in the adult suggest that operant conditioning does not require the MB (*Booker and Quinn, 1981*; *Wolf et al., 1998*; *Colomb and Brembs, 2010*; *Colomb and Brembs, 2016*) and may instead involve motor neuron plasticity (*Colomb and Brembs, 2016*). In classical conditioning of both adult and larval *Drosophila*, the MB has been identified as a convergence site for the external CS and the rewarding or punishing US (*Cognigni et al., 2018*; *Heisenberg et al., 1985*; *Heisenberg, 2003*; *Rohwedder et al., 2016*; *Vogt*

*et al., 2014*; *Saumweber et al., 2018*; *Owald and Waddell, 2015*). In each larval brain hemisphere, the MB comprises approximately 110 CS-encoding Kenyon cells (KCs) (*Aso et al., 2014*; *Honegger et al., 2011*; *Berck et al., 2016*; *Lin et al., 2014*; *Owald and Waddell, 2015*; *Campbell et al., 2013*; *Turner et al., 2008*; *Eichler et al., 2017*) that synapse onto 24 MB output neurons (MBONs) driving approach or avoidance (*Aso et al., 2014*; *Owald and Waddell, 2015*; *Perisse et al., 2016*; *Séjourné et al., 2011*; *Saumweber et al., 2018*; *Shyu et al., 2017*; *Plaçais et al., 2013*; *Eichler et al., 2017*). Dopaminergic and octopaminergic neurons that represent the rewarding or punishing US modulate KC to MBON connection strength (*Schwaerzel et al., 2003*; *Schroll et al., 2006*; *Honjo and Furukubo-Tokunaga, 2009*; *Vogt et al., 2014*; *Saumweber et al., 2018*; *Waddell, 2013*; *Eschbach et al., 2021*). The extent to which the MB is dispensable in larval operant conditioning is, by contrast, unknown.

We investigated whether subsets of Ddc neurons in the brain and SEZ could support memory formation in our bend direction paradigm. Gal80 under control of the tsh promoter suppresses expression in the VNC, but not in the brain or SEZ (*Clyne and Miesenböck, 2008*; *Figure 4—figure supplement 1*). We took an intersectional approach by targeting these transgenes with the LexA/LexAop binary system (*Simpson, 2016*) and expressing CsChrimson in Ddc neurons using Gal4/UAS. Prior to training with our operant conditioning protocol (*Figure 4B*), Ddc>CsChrimson, tsh >Gal80 larvae showed no directional bias in bend rate (p>0.5; *Figure 4E*). Observations of larval bend rates during training revealed a persistent, and in some cases statistically significant, direction preference to the unstimulated side (*Figure 4—figure supplement 2*). In vertebrates, dopamine release is crucial for not only learning but also action selection (*Grillner et al., 2013*). We wondered whether our targeted activation of only brain and SEZ dopaminergic neurons affected the larval motor program in a functionally analogous manner, causing the observed decrease in bends to the stimulated side. Following training, these larvae were equally likely to bend towards the side where they had previously received the optogenetic stimulus (mean = 3.55, sd = 2.33, n = 175) as they were to bend towards the previously unstimulated side (mean = 3.53, sd = 2.36, n = 175) (p>0.5; *Figure 4E*). The after training difference in bend rate for these larvae (mean = 0.029, sd = 2.41, n = 175) was significantly lower than that of Ddc>CsChrimson larvae (mean = 0.74, sd = 2.62, n = 143) (p=0.012, CLES = 0.58; *Figure 4H*). Based on these results, the dopaminergic and serotonergic neurons in the brain are not sufficient for operant conditioning. In contrast, the dopaminergic and serotonergic neurons in the VNC appear critical to the bend direction preference formed following paired optogenetic activation of all Ddc neurons.

We also tested whether exclusively activating the PAM cluster dopaminergic neurons innervating the MB could induce operant conditioning. 58E02-Gal4 drives expression in the majority of these neurons (*Rohwedder et al., 2016*). In the test period before training, 58E02>CsChrimson larvae did not exhibit a bend direction preference (p>0.5; *Figure 4F*). During training, however, these larvae showed a significant bend direction preference to the unstimulated side (*Figure 4—figure supplement 2*). Future work is necessary to assess whether these preference results are partly a consequence of these neurons' role in motor control. 58E02 neurons have synaptic connections to aversive MBONs (*Eichler et al., 2017*), though the absence of functional testing in larvae leads to uncertainty about whether these connections are excitatory, inhibitory, or modulatory. Following training, 58E02>CsChrimson larvae did not exhibit a learned direction preference for bends to either the previously stimulated (mean = 4.41, sd = 2.16, n = 145) or unstimulated (mean = 4.55, sd = 2.10, n = 145) side (p>0.5; *Figure 4F*). Because 58E02 comprises a small subset of Ddc neurons (for which fictive activation yielded an operant learning effect), these negative results also help confirm the absence of leaky UAS-CsChrimson expression. Indeed, these larvae exhibited a significantly lower difference in bend rate after training (mean = -0.14, sd = 2.71, n = 145) compared to Ddc>CsChrimson larvae (mean = 0.74, sd = 2.62, n = 143) (p=0.015, CLES = 0.58; *Figure 4H*). It remains to be seen whether these PAM cluster neurons contribute to memory formation by interacting with other Ddc neurons, especially those in the VNC. These results do, however, further support the idea that operant conditioning in *Drosophila* may not be mediated by the MB.

## Serotonergic VNC neurons may play role in operant conditioning of bend direction

Pairing an action with activation of numerous dopaminergic and serotonergic neurons across the CNS was sufficient to induce operant conditioning of bend direction. Furthermore, our results indicated

that the VNC subset of these neurons was critical to memory formation in the paradigm. It was an open question, however, whether this learning was mediated by dopamine, serotonin, or both. We expressed CsChrimson under the control of two sparse dopaminergic and serotonergic driver lines to investigate whether either neurotransmitter could exclusively induce operant conditioning of bend direction.

TH-Gal4 covers most dopaminergic neurons except the PAM cluster (*Rohwedder et al., 2016*). Under our high-throughput training protocol (*Figure 4B*), TH>CsChrimson larvae showed no difference in bend rate between the previously stimulated and unstimulated sides in the 1 min test periods either before (mean = 5.29, sd = 2.03 stimulated versus mean = 5.77, sd = 2.60 unstimulated; n = 75) or after (mean = 4.39, sd = 2.28 stimulated versus mean = 3.98, sd = 1.93 unstimulated; n = 122) training (p>0.5; *Figure 5A*). Tph-Gal4 targets the majority of serotonergic neurons and no dopaminergic neurons across the CNS of third-instar larvae (*Huser et al., 2012*). Before training, Tph>CsChrimson larvae showed no difference in bend rate between sides (mean = 5.27, sd = 3.53 stimulated versus mean = 4.86, sd = 2.83 unstimulated; n = 77) (p>0.5; *Figure 5B*). Paired activation of Tph neurons during bends to one side resulted in a significantly higher bend rate to the stimulated side (mean = 3.62, sd = 2.15, n = 126) relative to the unstimulated side (mean = 3.29, sd = 2.46, n = 126) after training (p=0.019, CLES = 0.58; *Figure 5B*). Notably, no statistically significant difference was observed when directly comparing the difference in bend rate between TH>CsChrimson (mean = 0.41, sd = 2.42, n = 122) and Tph>CsChrimson (mean = 0.33, sd = 2.51, n = 126) larvae after training (p>0.5; *Figure 5D*). Together, these results suggest a cautious interpretation. Although activation of Tph serotonergic neurons on their own appears sufficient to form a learned direction preference in our paradigm, we cannot exclude the possibility that TH dopamine neurons also contribute to operant learning, even though they were not sufficient to induce learning.

Based on our finding that operant conditioning failed following restriction of Ddc >CsChrimson expression to the brain and SEZ, we wondered whether serotonergic neurons in the VNC were also critical for operant memory formation in this paradigm. We used tsh-Gal80 to restrict the Tph-Gal4 expression pattern to the brain and SEZ. Tph>CsChrimson, tsh>Gal80 larvae showed no significant direction preference prior to training (mean = 3.46, sd = 2.52 stimulated versus mean = 3.99, sd = 2.97 unstimulated; n = 76) (p>0.5; *Figure 5C*). Paired optogenetic activation of brain and SEZ Tph neurons with larval bends to one side did not induce a learned direction preference (mean = 1.86, sd = 1.70 stimulated versus mean = 1.98, sd = 1.74 unstimulated; n = 111) (p>0.05; *Figure 5C*). The difference in bend rate after training for Tph>CsChrimson, tsh>Gal80 larvae (mean = -0.12, sd = 1.83, n = 111) was significantly lower than that of Tph>CsChrimson larvae (mean = 0.33, sd = 2.51, n = 126) (p=0.025, CLES = 0.58; *Figure 5D*). The Tph-Gal4 expression pattern contains two neurons per VNC hemisegment (with the exception of a single neuron in each A8 abdominal hemisegment) (*Huser et al., 2012*). Future experiments exclusively targeting a single serotonergic neuron per VNC hemisegment could be valuable in determining whether they are sufficient for operant learning.

## Discussion

### Automated trace conditioning of *Drosophila* larvae

Our classical conditioning experiments showed that *Drosophila* larvae significantly reduce their attraction to Or42b after this fictive odour is repeatedly paired with Basin activation. This reduction in attraction was significantly greater than that observed when Or42b stimulation was repeatedly paired with heating alone. Such results build upon prior work using similar training methods (e.g. *Honda et al., 2014*), in particular by increasing the temporal precision of larval heating. These results demonstrate the efficacy of our larval tracking system for high-throughput automated classical conditioning, and show that larvae can form associations between purely fictive stimuli (i.e. delivered through independent opto- and thermogenetic stimulation). Most notably, we demonstrate for the first time that *Drosophila* larvae can perform classical conditioning in the absence of temporal overlap between CS and US (i.e. with more than a 5 s gap from CS offset to US onset). These observations extend the known limits of larval learning, and were made possible only by using a tracker with temporally precise opto- and thermogenetic stimulation.

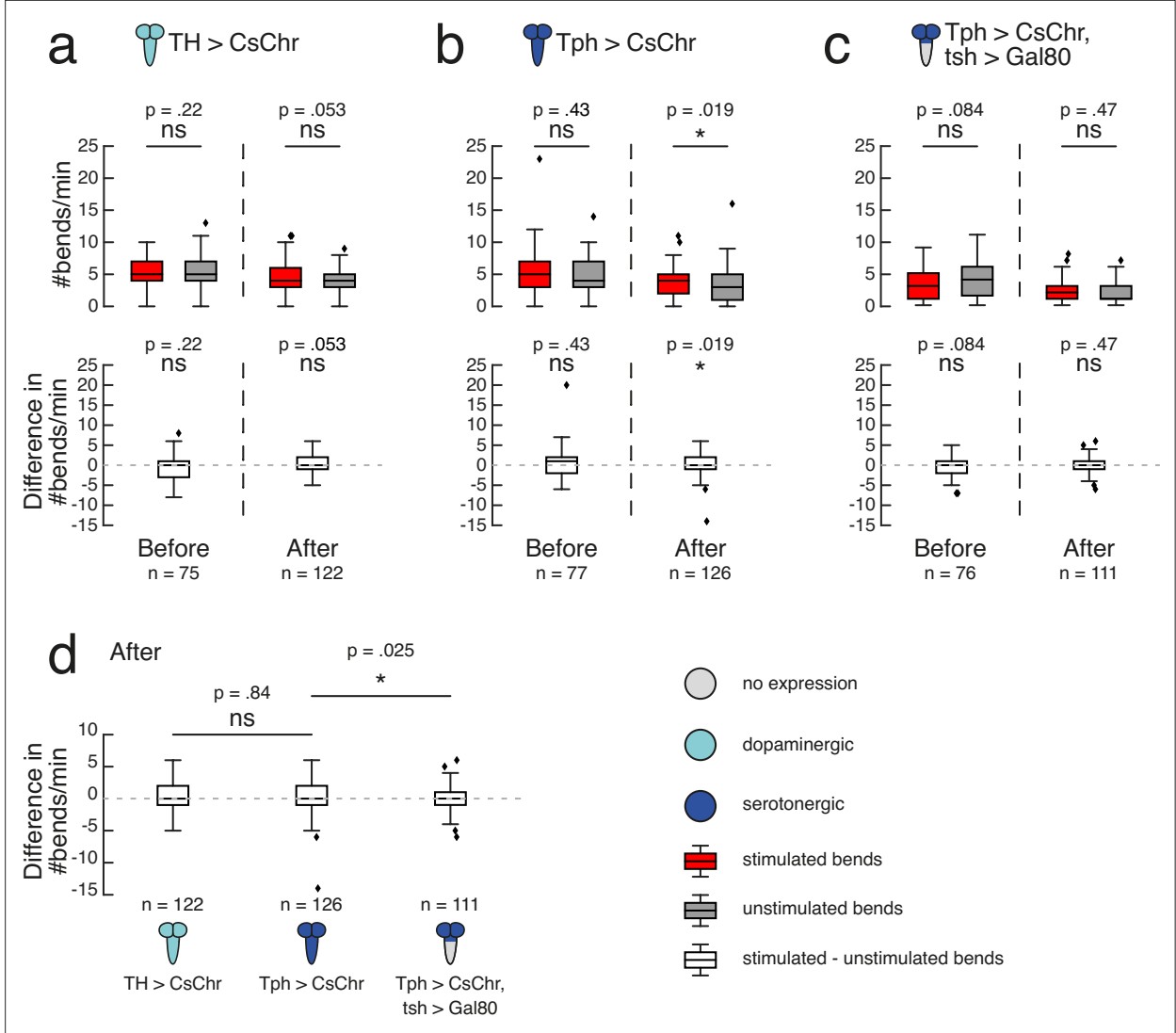

**Figure 5.** Serotonergic neurons may mediate operant conditioning. High-throughput experiments followed the protocol depicted in *Figure 4B*. Gal4 expression is depicted as color-coded CNS (see legend). UAS-CsChrimson effector abbreviated as CsChr for visual clarity. Bars show medians, as well as upper and lower quartiles of data. Outliers (filled diamonds) are randomly jittered horizontally to aid visualisation. (a–c) top row. Larval bend rate shown as number of bends per minute, grouped by bends to stimulated side (dark red) or unstimulated side (grey). Statistical comparisons calculated using a paired, two-sided Wilcoxon signed-rank test. (a–c) bottom row. Difference in bend rate (black) shown between the stimulated and unstimulated sides. Statistical comparisons calculated using a two-sided Wilcoxon signed-rank test. (a–c). Data shown from the test periods before training round 1 (Before) and after training round 4 (After). n is the number of larvae in each time bin. Exact p-values written above corresponding data. ns p ≥ .05 (not significant), * p < .05. (d) Difference in bend rate after training round 4 (same data as in a–c bottom row). Statistical comparisons against Tph > CsChr calculated with a two-sided Mann-Whitney *U* test, with Bonferroni correction; ns p ≥ .05/2 (not significant), * p < .05/2. See *Figure 5—source data 1*.

The online version of this article includes the following source data and figure supplement(s) for figure 5:

**Source data 1.** Source data showing that serotonergic neurons may mediate operant conditioning.

**Figure supplement 1.** Tph-Gal4 expression pattern without and with tsh-Gal80 restriction.

Future studies may wish to focus on the known ability of *Drosophila* larvae to perform time-dependent valence reversal learning. In this paradigm, the learned valence of a CS reverses when it is delivered before vs. after the US (*Saumweber et al., 2018*). While we observed reduced larval attraction to Or42b after forward-paired training, we did not observe increased attraction after backward-paired training. One explanation could be the slow temporal dynamics of larval cooling (*Figure 2—figure supplement 1*). It is possible that residual larval body heat after IR stimulation caused Basin neurons to remain partially active during the subsequent Or42b stimulation. Future

studies could address this limitation by using a backward-paired training protocol with optogenetic activation of Basin neurons and thermogenetic activation of Or42b. Following our novel observation of trace conditioning in larvae, future work could also investigate the maximum temporal gap between sensory stimuli over which larvae can still be conditioned. It will also be important to determine whether conditioning with a fictive odour stimulus changes larval responses to natural odours. For example, larvae showing reduced attraction to Or42b activation may show reduced attraction to natural odours that activate Or42b neurons (e.g. ethyl acetate).

## Automated operant conditioning of *Drosophila* larvae

We have also shown that *Drosophila* larvae are capable of operant conditioning via optogenetic activation of Ddc neurons. A significant increase in bending towards the previously stimulated side relative to the unstimulated side was only observed after coincident, positive reinforcement of this behaviour during training; a hallmark of operant conditioning. Because Ddc-Gal4 drives expression in dopaminergic and serotonergic neurons (*Li et al., 2000*; *Sitaraman et al., 2008*), we investigated which of these neurotransmitters support operant learning in our behavioural paradigm. Fictive activation of dopaminergic TH neurons paired with bends to one side did not yield a significant learned direction preference. In contrast, activation of serotonergic Tph neurons was sufficient to bias more bends towards the previously stimulated side. However, a direct comparison of differences in bend rate between the Tph and TH groups was itself not significant, raising the possibility that TH could weakly contribute to operant learning, even though it was not sufficient to induce significant learning in these experiments.

Of significant contrast to the Tph result was impaired operant conditioning after restricting Tph expression to only the brain and SEZ. These results suggest a novel role of VNC serotonergic neurons in conveying a reward signal for operant conditioning in *Drosophila* larvae.

It is noteworthy that *Nuwal et al., 2012* used optogenetic activation of sugar-sensing neurons to establish a walking direction preference in adult *Drosophila*. Although we were unsuccessful in using a similar US to condition larval bend direction (data not shown), our Ddc and Tph results suggest that other sensory rewards may mediate operant conditioning if the information retrieval occurs within the VNC itself. Investigating *FoxP* and protein kinase C (PKC) expression in larval neurons may aid in identifying candidate neuronal populations, given both genes' involvement in operant self-learning in adult *Drosophila* (*Mendoza et al., 2014*; *Brembs and Plendl, 2008*; *Colomb and Brembs, 2016*). Furthermore, developing sparser lines that target single serotonergic and dopaminergic neuron types will enable the identification of the smallest subsets of neurons that are sufficient for providing the operant learning signal. Behavioural experiments with these genetic lines may have the added benefit of mitigating conflicting or non-specific reinforcement signalling.

There are limitations to interpreting behaviour following fictive neuronal activation. It will therefore be important to replicate the new paradigms developed with this rig using lower-throughput methods with natural stimuli. For example, one might modify our multi-larva tracker to accommodate closed-loop presentation of real, innately appetitive gustatory stimuli by using closed-loop control of microfluidic devices. One could also try conditioning behaviours or behavioural sequences for which operant self-learning may be more ethologically relevant (e. g. individual or cooperative digging). Importantly, however, the frequency of such behaviour in naïve, freely behaving larvae may affect the amount of US received during paired training, making observable memory formation more difficult. Our multi-larva tracker could potentially address this challenge with probabilistic, thermogenetic activation of associated command neurons and optogenetic reward when performing the desired action.

## Multi-larva tracker is a tool with wide applicability

Due to available genetic tools and the emerging connectome, the *Drosophila* larva is a uniquely advantageous model organism for neuroscience. Knowing this, we built a system that combines

FPGA-based real-time tracking of multiple, freely-behaving larvae with independent control of larval illumination and heating. Delivering these external stimuli to larvae expressing opto- and thermosensitive proteins in their CNS allows precise activation of genetically-targeted neuronal populations. We also developed robust online behaviour detection, enabling modulated stimulus presentation as a function of individual larval action. This approach significantly broadens the range of sensory information that can be delivered in laboratory experiments, unconstrained by the need to deliver real odours, tastants, or mechanosensory experiences. A notable limitation of the existing software architecture is an inability to maintain object identity following larval collisions with each other or the plate's edge. This could be addressed in future software modifications.

In summary, the work we presented here demonstrates the utility of our FPGA-based high-throughput, tracking and training system for developing novel classical and operant conditioning paradigms in *Drosophila* larvae. This system will enable rapid screening for neurons and circuits that underpin different forms of learning and also has the potential to drive future research in larval taxis (*Luo et al., 2010*; *Gomez-Marin et al., 2011*; *Kane et al., 2013*; *Jovanic et al., 2019*), decision-making (*Eschbach et al., 2021*; *Krajbich, 2019*; *DasGupta et al., 2014*), and spatial navigation and memory (*Neuser et al., 2008*; *Haberkern et al., 2019*).

# Materials and methods

**Key resources table**

| Reagent type (species) or resource | Designation | Source or reference | Identifiers | Additional information |
|---|---|---|---|---|
| Genetic reagent (*D. melanogaster*) | w[1118]; P{y[+t7.7] w[+mC]= GMR72 F11-Gal4}attP2 (72F11-Gal4) | Bloomington Stock Center | RRID:BDSC_39786 | |
| Genetic reagent (*D. melanogaster*) | w[1118]; P{y[+t7.7] w[+mC]= GMR69 F06-GAL4}attP2 (69F06-Gal4) | Bloomington Stock Center | RRID:BDSC_39497 | |
| Genetic reagent (*D. melanogaster*) | w[1118];; attP2 | *Pfeiffer et al., 2008* | | |
| Genetic reagent (*D. melanogaster*) | w[1118] P{y[+t7.7] w[+mC]= 20XUAS-IVS-CsChrimson. mVenus}attP18 (UAS-CsChrimson) | Bloomington Stock Center | RRID:BDSC_55134 | |
| Genetic reagent (*D. melanogaster*) | UAS-dTrpA1 | Dr Paul Garrity | | |
| Genetic reagent (*D. melanogaster*) | 13XLexAop2-CsChrimson-tdTomato in attP18; Or42b-LexAp65 in JK22C; + (Or42b>CsChrimson) | Janelia Research Campus | | |
| Genetic reagent (*D. melanogaster*) | w; UAS-dTRPA1, 13XLexAop2-GCAMP6s 50.641 in Su(Hw)attP5 (/Cyo); 72F11-GAL4 in attP2 (72F11>dTrpA1) | Bloomington Stock Center | RRID:BDSC_44590 | |
| Genetic reagent (*D. melanogaster*) | w[1118];; Ddc-Gal4-HL8-3D (Ddc-Gal4) | *Li et al., 2000* | | |
| Genetic reagent (*D. melanogaster*) | w[1118]; P{y[+t7.7] w[+mC]= GMR58E02-GAL4}attP2 (58E02-Gal4) | Bloomington Stock Center | RRID:BDSC_41347 | |
| Genetic reagent (*D. melanogaster*) | TH-Gal4 | Bloomington Stock Center | RRID:BDSC_8848 | |
| Genetic reagent (*D. melanogaster*) | +; Tph-Gal4; + | *Park et al., 2006* | | |
| Genetic reagent (*D. melanogaster*) | 20xUAS-CsChrimson-mVenus@attP18; tsh-LexA, pJFRC20-8xLexAop2-IVS-Gal80-WPRE (su(Hw)attP5)/CyO, 2xTB- RFP; + (UAS-CsChrimson; tsh-LexA, LexAop-Gal80) | Dr Stefan Pulver, Dr Yoshinori Aso | | |
| Genetic reagent (*D. melanogaster*) | 10XUAS-IVS-myr::smGFP-HA@attP18, 13XLexAop2-IVS-myr::smGFP-V5@su(Hw)attP8 (UAS-GFP) | *Nern et al., 2015* | | |

*Continued on next page*

*Continued*

| Reagent type (species) or resource | Designation | Source or reference | Identifiers | Additional information |
|---|---|---|---|---|
| Software, algorithm | Custom scripts to run closed-loop multi-larva tracking and opto-/thermo-genetic stimulation | https://github.com/ZlaticLab/multi-larva-tracker-scripts-public; *Clayton et al., 2022* | | |

## Multi-larva tracker

### Hardware setup

A high-resolution camera (3072x3,200 pixels) (#TEL-G3-CM10-M5105, Teledyne DALSA, Ontario, Canada) positioned above a 23 cm x 23 cm 4% agarose plate captured 8-bit greyscale images at 20 Hz. The agarose plate was illuminated from below by a 30 cm x 30 cm 850 nm LED backlight (#SOBL-300x300–850, Smart Vision Lights, Norton Shores, Michigan) equipped with intensity control (#IVP-C1, Smart Vision Lights, Norton Shores, Michigan). An 800 nm longpass filter (#LP800-40.5, Midwest Optical Systems, Palatine, Illinois) mounted on the camera blocked all visible wavelengths, including those used for optogenetics. When the agarose plate comprised most of the camera image, each pixel corresponded to either 72.92 µm (for proof-of-principle and operant conditioning experiments) or 75.84 µm (for temporal dynamics of larval heating and classical conditioning experiments) (*Figure 1*).

Each camera image was processed in parallel on both the host computer (#T7920, running Windows 10, Dell Technologies Inc, Round Rock, Texas) and an FPGA device (#PCIe-1473R-LX110, National Instruments, Austin, Texas). LabVIEW 2017 (National Instruments, Austin, Texas) software extracted larval contours and interfaced with C++software that performed real-time behaviour detection. The LabVIEW software controlled closed-loop optogenetic and thermogenetic stimulation in response to these detected behaviours (*Figure 1*). All relevant experiment parameters and time-series data were output for offline analysis through a custom MATLAB framework (see LABEL:sec:materials_and_methods).

### Multi-animal detection and tracking

Raw camera images were read by the FPGA at 20 Hz and then sent to the host computer. The LabVIEW process on the host computer then filtered out non-larval objects by combining background subtraction and binary thresholding. The remaining objects were each enclosed in a rectangular box of minimal size, with edges parallel to the camera image axes (*Figure 1D*). We defined the following criteria to detect third-instar larvae within these boxes:

- Pixel intensity range (default 25–170): the minimum and maximum brightness values for pixels selected by binary thresholding (between 0 and 255 for an 8-bit image).
- Box side length (pixels) (default 6–100): the range of eligible values for width and height of each box.
- Box width +height (pixels) (default 12–200): the range of eligible values for the sum of each box's width and height.
- Box area (pixels) (default 300–900): the range of eligible values for the area of each box.

To track larvae over time, the host computer assigned a numerical identifier to each eligible object. We used distance-based tracking with a hard threshold of 40 pixels to maintain larval ID based on centroid position. Although identity was lost when larvae touched or reached the plate's edge, new IDs were generated when larvae matched detection criteria. For each of the largest 16 objects, the host computer sent a binary pixel pattern and location (defined as the centre of the box) to the FPGA. Since the host computer required more than 50ms of run time for object detection, this process was not executed in every frame. On average, the FPGA received updated objects and their locations every three frames.

The FPGA extracted object contours in three steps. Within a 2 cm$^2$ region of interest around the object's centre, the FPGA first applied a user-defined binary threshold, then applied both vertical and a horizontal convolution with a 2x1XOR kernel, and finally generated edge pixels by combining the results of the two convolutions using an OR operation. Contours were extracted from edge images using the Moore boundary tracing algorithm (*Gonzalez and Woods, 2018*) with three added error

capture procedures. First, if the algorithm yielded a contour that ended prematurely or contained small loops, the construction process could be reversed by up to 16 contour points to find an alternative contour. Second, 10,000 FPGA clock cycles ($\approx$ 100 us) was the maximum allotted execution time, with each pixel comparison occurring within one clock cycle. In the rare event that this window was exceeded, the algorithm returned the already constructed contour points. Third, a contour containing fewer than 63 points was rejected and the FPGA returned the last valid contour detected for a given larva ID. The algorithm stopped when none of the remaining neighbours were edge pixels (*Figure 1—figure supplement 1*).

## Contour processing and landmark detection

An undesired result of the FPGA contouring algorithm was the variable number of contour points across larvae and frames. We aimed to detect behaviour based on a smooth contour with a fixed number of 100 contour points. This contour regularization was achieved inside the Behaviour Programme using Fourier decomposition and reconstruction as in *Masson et al., 2020*.

The initial detection of head and tail was implemented on FPGA. The larva's head and tail were defined as the contour points with the sharpest and second-sharpest curvature, respectively (*Figure 1—figure supplement 2*). While correct in most cases, this calculation sometimes led to flipped detection of the two body ends. The Behaviour Programme flagged and corrected these false detection events at run time by calculating the distance head and tail traveled between frames and tracking the number of correct versus flipped detection events. The vote system correction commonly failed when the larva made large angle bends. The resulting contour was nearly-circular and exhibited similar curvature across all points. The solution required resetting the vote tallies when detecting these ball events (*Figure 1—figure supplement 2*).

We defined the larval spine as 11 points running along the central body axis from head to tail (*Figure 1—figure supplement 3*; *Swierczek et al., 2011*). In addition to head and tail, the Behaviour Programme calculated three equally distributed landmark points along the spine (neck_top, neck, and neck_down). A fourth landmark, the centroid, defined the larva's location. The six landmarks were collectively used to extract features for training behaviour classifiers (*Figure 1—figure supplement 3*).

The Behaviour Programme transformed the raw contour and spine from camera coordinates (in pixels) to world coordinates (in mm). If stable larval detection criteria were met, all spine points were temporally smoothed using exponential smoothing (*Figure 1—figure supplement 3*).

## Feature extraction

We developed a machine learning approach to address the high deformability of the larva shape, ensure live execution, reduce overfitting, and limit the volume of data tagging. What follows is a brief summary of larval features describing motion direction, body shape, and velocity that were calculated from the contour and spine data inside the Behaviour Programme. Features were designed as in *Masson et al., 2020*, with notable modifications required to run the inference live:

1. Motion Direction (*Figure 1—figure supplement 4*)
   `direction_vector`: normalised vector describing the main body axis
   `direction_head_vector`: normalised vector describing the head axis
   `direction_tail_vector`: normalised vector describing the tail axis
2. Body Shape (*Figure 1—figure supplement 5*)
   `skeleton_length`: summed distances between consecutive spine points
   `perimeter`: summed distances between neighbouring contour points
   `larva_arc_ratio`: ratio of contour perimeter to convex hull perimeter (`larva_arc_ratio` $\geq 1$ and was close to 1 when larva was in either straight or ball-like shape)
   `larva_area_ratio`: ratio of the areas enclosed by the contour and its convex hull ($0 \leq$ `larva_area_ratio` $\leq 1$ and was close to 1 when the larva was in either straight, heavily curved, or ball-like shape)
   `eig_reduced`: `eig_reduced` $= \frac{|\lambda_1 - \lambda_2|}{\lambda_1 + \lambda_2}$ where $\lambda_1, \lambda_2$ were the eigenvalues of the structure tensor of the larval contour with respect to the neck ($0 \leq$ `eig_reduced` $\leq 1$ and eig_reduced decreased as the bend amplitude of the larva increased)

`s`: normalised angle along the body ($-0.5 \leq s \leq 1$, was close to 1 when larva was straight, and decreased with increasing bend amplitude)

`asymmetry`: sine of the angle between direction_vector and direction_head_vector (`asymmetry` > 0 when larva bent left and `asymmetry` < 0 when larva bent right)

`angle_upper_lower`: absolute angle between direction_vector and direction_head_vector (despite similarity to asymmetry, this develops different dynamics following temporal smoothing, which are valuable for stable left and right bend detection)

3. Velocity (*Figure 1—figure supplement 6*)

Velocity of all six landmark points (head_speed, neck_top_speed, neck_speed, neck_down_ speed, tail_speed, and v_centroid) in mm/s over interval $dt = 0.2\,s$ (four frames)

`v_norm`: arithmetic mean of neck_top_speed, neck_speed, and neck_down_speed, passed through a hyperbolic tangent activation function to suppress excessively large values

`speed_reduced`: relative contribution of neck_top_speed to v_norm, passed through a hyperbolic tangent activation function to suppress excessively large values (speed_reduced increased when the anterior larval body moved quickly compared to the posterior, e. g. when a bend was initiated)

`damped_distance`: distance (mm) travelled by neck, giving greater weight to recent over past events

`crab_speed`: lateral velocity (mm/s), defined as the component of neck_speed orthogonal to direction_vector_filtered

`parallel_speed`: forward velocity (mm/s), defined as the component of neck_speed_filtered parallel to direction_vector_filtered

`parallel_speed_tail_raw`: tail's forward velocity (mm/s), defined as the component of tail_ speed_filtered parallel to direction_tail_vector_filtered

`parallel_speed_tail`: similar to parallel_speed_tail_raw, with the difference that tail_ speed_filtered was normalised prior to calculating the dot product (i. e. a measure of tail movement direction which took values between –1 (backward) and+1 (forward))

To extract features in real time and address various sources of noise, we implemented exponential smoothing defined as follows for a given feature f (*Figure 1—figure supplement 7*).

$$\texttt{f\_filtered}_t = (1 - \alpha) \cdot \texttt{f\_filtered}_{t-\Delta t} + \alpha \cdot \texttt{f}_t$$

where t is unitless, but derived from the experiment time in seconds, $\alpha = \frac{\Delta t}{\tau}$ with $\Delta t = 0.05\,s$ and $\tau = 0.25\,s$. Features that had the potential to exhibit large value deviations (e. g. v_norm) were instead bounded using a hyperbolic tangent function. Additionally, some features were exponentially smoothed over a longer time window (where $\alpha_{long} = \frac{\Delta t}{\tau_{long}}$ with $\Delta t = 0.05\,s$ and $\tau_{long} = 5\,s$) (*Figure 1— figure supplement 7*).

Convolution was used to approximate a smoothed squared derivative for each feature (*Figure 1— figure supplement 8*); useful for integrating information over time without needing to further expand the feature space. The underlying mathematical concepts were motivated by *Masson et al., 2012*. For a given feature f at time $t$, f_convolved_squared was calculated as follows:

$$f1_t = (1 - \lambda \Delta t).f1_{t-\Delta t} + \tfrac{1}{2}\Delta t.(f_{t-\Delta t} + f_t)$$
$$f2_t = \lambda \Delta t.f1_{t-\Delta t} + (1 - \lambda \Delta t).f2_{t-n\Delta t}$$
$$\texttt{f\_convolved\_squared}_t = k.(f1_t - f2_t)^2,$$

where $\Delta t = 0.05s$, $\lambda = \frac{1}{\tau}$, $\tau = 0.25s$, and n=5s, k values were empirically chosen for each feature.

## Behaviour classifiers

Behaviour classifiers were developed using a user interface similar to JAABA (*Kabra et al., 2013*). The underlying algorithms combined trained neural networks and empirically-determined linear thresholds. We developed a MATLAB (MathWorks, Natick, Massachusetts) user interface with functions for data visualisation, manual annotation, and machine learning using the Neural Network Toolbox, the Deep Learning Toolbox, and the Statistics and Machine Learning Toolbox. Here, we briefly describe the behaviour classifiers and provide performance results based on manual validation (*Table 1*).

**Table 1.** Manual quantification of behaviour detection performance.

| back (268 events from 24 larvae in 60min of video data) | |
|---|---|
| Precision | 86.5% |
| Recall | 88.4% |
| **bend (714 events from 24 larvae in 60 min of video data)** | |
| Precision | 95.6% |
| Recall | 96.4% |
| Accuracy of left and right detection (true-positive bends) | 97.3% |
| **forward (425 events from 24 larvae in 60 min of video data)** | |
| Precision | 97.8% |
| Recall | 94.1% |
| **forward_peristaltic (2954 events from 24 larvae in 60 min of video data)** | |
| Precision | 99.5% |
| Recall | 93.6% |
| Events which are falsely combined with another event | 10.7% |
| Events which are detected as more than one event | 1.2% |
| **roll (240 events from 24 larvae in 60 min of video data)** | |
| Precision (rolls and roll-like events) | 96.6% |
| Recall (rolls) | 86.7% |
| Recall (roll-like events) | 25.8% |

The bend classifier was based on predefined thresholds for temporally smoothed body shape features and was itself exponentially smoothed over time. Independent left and right classifiers were used to initially detect bend direction. To detect left and right bends, these classifiers were combined with the smoothed bend classifier using an AND conjunction.

To improve left and right detection performance, we developed a classifier for circular larval contours. This ball classifier used a feed-forward neural network with a single fully connected hidden layer whose inputs were normalised values of `eig_reduced`, `larva_arc_ratio`, and `larva_area_ratio`. The hidden layer consisted of five neurons with a hyperbolic tangent activation function. The output layer contained a single neuron and used a sigmoid activation function. The neural network was trained in MATLAB on a manually annotated data set for 500 epochs using a cross-entropy loss function and scaled conjugate gradient backpropagation. If a ball was detected within the previous 1.5 s, left and right classifiers were overwritten to match the last detected bend direction prior to the beginning of the ball.

The back classifier detected individual backward peristaltic waves based on thresholds for smoothed tail velocity features combined with no ball detection within the previous 1.5s.

Two different classifiers were used to detect crawling. forward detected longer forward crawl periods based on thresholds for smoothed tail velocity features combined with no ball detection within the previous 1.5 s. `forward_peristaltic` detected individual forward peristaltic waves based on the forward classifier and a threshold on forward tail velocity.

The roll classifier was based on thresholds for body shape and velocity combined with no ball detection and was exponentially smoothed over time. If a roll was detected within the previous 1.5 s, forward, `forward_peristaltic`, and back classifier values were reset to reduce false-positive detection for these classifiers. Unusual behaviour patterns such as rapid bending or twitching could be observed in addition to true larval rolling. These behaviours were considered 'roll-like' events during manual validation of the roll classifier's performance.

## Optogenetic stimulation

Optogenetic stimulation was achieved using two digital micromirror devices DMDs to project light patterns onto larvae on the agarose plate (*Figure 2C*). Both DMDs operated like a monochrome red light projector (1920x1080 pixel in classical conditioning experiments, 768x1024 pixel in proof-of-principle and operant conditioning experiments) with numerous rotatable micromirrors used to modulate the intensity of individual pixels. Because each DMD on its own was insufficient for optogenetic stimulation of larvae, we installed both devices on the system with their projections each covering the entire agarose plate (*Figure 1B*). In this way, the summed light intensities of the two DMDs could be achieved at all locations. For the proof-of-principle and operant conditioning experiments, one DMD contained an integrated 613 nm LED (#CEL-5500-LED, Digital Light Innovations, Austin, Texas) and the other (#CEL-5500-FIBER, Digital Light Innovations, Austin, Texas) received input from an external 625 nm LED (#BLS-GCS-0625–38A0710, Mightex Systems, Ontario, Canada) controlled by a BioLED light source control module (#BLS-13000–1, Mightex Systems, Ontario, Canada) and fed through an optic fibre (#LLG-05-59-420-2000-1, Mightex Systems, Ontario, Canada) (*Figure 1C*). For the classical conditioning experiments, both DMDs (#DLP4710EVM-LC, Texas Instruments) each received input from an external 625 nm LED source (#LE A P1W-RSSP-23, Osram, Germany).

Accurately aiming light at crawling larvae required spatial calibration of each DMD. Calibration was performed by projecting square spots at fixed DMD pixel locations and linearly fitting the corresponding camera coordinates. We also determined that DMD illumination using the default light output was not uniform at plate level, which could have resulted in variable optogenetic stimulation depending on larval location. We therefore normalised the pixel intensity of the DMD image to the highest intensity uniformly achievable at all plate locations. A look-up table containing the normalisation factor for each DMD pixel was then calculated using bi-linear interpolation with approximately 100 light intensity values measured across the plate. To accommodate for possible differences in non-uniformity between the two DMDs, this intensity calibration was performed for both DMDs simultaneously following spatial calibration.

A user-defined Behaviour Programme protocol operated on the behaviour detection output and sent 8-bit optogenetic stimulation instructions to the LabVIEW application. Because the LabVIEW application updated DMD projections at 20 Hz, the delay between behaviour detection and closed-loop optogenetic stimulation of individual larvae did not exceed 50ms (*Figure 1D*). Furthermore, if two or more larvae were close enough such that their corresponding stimulation areas overlapped, the light intensity in the overlapping region was set to the smallest of those values to avoid undesired stimulation.

## Thermogenetic stimulation

Thermogenetic stimulation for proof-of-principle experiments was achieved by heating up larvae with a custom-built 1490 nm IR laser setup (#2CM-101, SemiNex, Peabody, Massachusetts). An out-of-the-box 1470 nm IR laser was used for temporal dynamics of larval heating and in our classical conditioning experiments (#LRD-1470-PFI-15000–05, Laserglow Technologies, Ontario, Canada). Each laser's light guide was fed into a two-axis galvanometer system (#GVSM002, Thorlabs, Newton, New Jersey; *Figure 2B*). Both the laser and its corresponding galvanometer were controlled by an analogue output device (#PCIe-6738, National Instruments, Austin, Texas; *Figure 1C*). Two mirrors inside the galvanometer were rotated around orthogonal axes to target the laser beam spot to any user-defined location on the agarose plate. Mirror positions were controlled by two integrated motors receiving location-specific voltage inputs. The beam spots for the 1470 nm and 1490 nm lasers measured approximately 2.3 cm and 5 mm in diameter, respectively, depending on each beam's angle of incidence to the plate.

Spatially calibrating the galvanometer was necessary to obtain a map between larval locations in world coordinates and the mirror motor input voltages. A visible aiming beam was scanned across the agarose plate using a fixed set of voltage pair inputs to the galvanometer. With the optical filter removed from the camera, the aiming beam's location in camera coordinates was automatically extracted from the image using binary thresholding. Two voltage-to-camera look-up tables were generated through bi-linear interpolation of these measured coordinates. For accurately targeted thermogenetic stimulation, the location of the larval centroid was first converted to camera coordinates using the existing

world-to-camera transform and was then mapped to a pair of galvanometer input voltages using the look-up tables.

When designing our tracker's thermogenetic stimulation system, we were careful to ensure that all larvae received the same stimulation regardless of their position on the agarose plate. A larva's location changed the laser beam's angle of incidence, causing the illuminated spot at plate level to take an elliptical shape with variable size. If we were to keep laser beam power constant, the changing spot area would generate inconsistencies in the intensity of IR light projected over each larva. For proof-of-principle experiments, laser intensity calibration was used to normalise the 1490 nm laser intensity to achieve constant power per unit area. Prior to any experimentation, the laser's visible aiming beam was scanned across the plate and the camera image was used to measure the beam's spot size at various locations. Bi-linear interpolation was then used to generate a pixel-wise look-up table containing the laser power scaling factors. We also accounted for a nonlinear relationship between the 1490 nm laser source input voltage and the total power output by generating a voltage-to-power map from manual measurements. With these transformations, the system could calculate the laser source input voltage necessary to produce uniform stimulation at any location. For temporal dynamics of larval heating and classical conditioning experiments, we upgraded our tracker with a heat camera (Teledyne FLIR AX8) for real-time measurement and maintenance of larval body temperature. The heat camera captured thermal images of the agarose plate at 10 Hz and sent them to the host computer. The host computer used this information together with behaviour camera data to extract temperature readings at each larval location (*Figure 1D*). A custom closed-loop software system (LabVIEW and Python) then iteratively updated the 1470 nm IR laser intensity *I* at each larval location using the following sigmoid function:

$$I = \frac{100}{1+e^{\frac{x}{2}}}$$

in which I is a percentage of the maximum laser output (15 W) and x is the current recorded temperature minus the target temperature in °C.

A user-defined Behaviour Programme protocol operated on the 20 Hz behaviour detection output and sent thermogenetic stimulation instructions to the LabVIEW application which controlled the galvanometer and laser (*Figure 1D*). Larval centroid locations were specified on every frame, enabling a single galvanometer to cycle the laser beam between all individual larvae at 20 Hz. All larvae were stimulated within the available 50ms time window (i. e. 11ms per larva with four larvae on the plate, but 5.5ms per larva with eight larvae on the plate). Switching off the laser input for 1.5ms between larvae accounted for small time fluctuations surrounding each new galvanometer position update and helped to avoid undesired stimulation of other plate areas (*Figure 2B*). If fewer objects were detected in a given frame than expected, the remaining galvanometer target locations were set to the plate's centre and the corresponding laser intensity was set to zero. This temporal pattern of galvanometer position updates yielded no more than 100ms delay between behaviour detection and closed-loop thermogenetic stimulation.

## Single-larva tracker
### Hardware setup and software framework

The single-larva tracker (*Figure 4—figure supplement 3*) electronics and hardware were nearly identical to the tracker described in *Schulze et al., 2015*, with notable exceptions to the camera and backlight which are detailed here. The behaviour arena comprised a layer of 1% agarose sitting atop a fixed glass plate, prepared daily to maintain moisture. Above and below the arena sat two motorised linear slides (#T-LSR450B, Zaber Technologies), arranged perpendicular to one another so that mounted components could move to any (x,y) location within the 34 cm x 38 cm coverage area. The host computer sent 4 Hz (x,y) position updates to the linear slides. Mounted to the slides above the behaviour arena was a high-resolution camera (2048x2048 pixels) (Grasshopper3 #GS3-U3-41C6NIR-C, Point Grey Research) that captured 20 Hz images and sent them to the host computer for real-time display on a graphical user interface. A long-distance microscope (Model KC/S VideoMax with IF2 objective, Edmund Optics) was c-mounted to the camera. Also mounted to the slides above the behaviour arena was a 617 nm LED (#PLS-0617–030S, Mightex Systems), with a 10 Hz update rate controlled by a universal driver (#SLC-xx04-US, Mightex Systems). The centre of the LED projection

on the agarose was aligned with the centre of the camera image to facilitate targeted optogenetic excitation. To maintain the integrity of behaviour detection, stimulus presentation, and optogenetic experimentation, all hardware was housed inside a light-tight enclosure. The agarose plate was illuminated from below by a 2 in x 2 in 880 nm LED backlight (#BL0202-880IC, Advanced Illumination) mounted to the linear slides. The hardware communication and operation software was written under a Robot Operating System (ROS) framework.

## Contour processing, behaviour detection, and optogenetic stimulation

The host computer software extracted the larval contour by applying inverse binary thresholding to each raw camera image. As with the multi-larva tracker, contour regularization was achieved using Fourier decomposition and reconstruction. Also the same as the multi-larva tracker, the single-larva tracker software detected the larva's head and tail using the contour's sharpest and second-sharpest internal angles, respectively, and corrected errors in real time using proximity measurements and a vote system (for algorithm details, see section above on multi-larva tracker contour processing and also *Figure 4—figure supplement 2*). The software calculated three other landmarks at equally distributed points along the larval spine. The neck landmark defined the larva's location and was used to update the linear slide positions, keeping the larva centred in the camera's field of view (*Figure 4—figure supplement 3*). As in the multi-larva tracker software, larval features describing motion direction, body shape, and velocity were calculated frame-by-frame from the contour and spine data and processed to remove high-frequency noise.

The left and right bend classifier consisted of a neural network with a single fully connected hidden layer whose inputs were the body shape features s and `eig_reduced`, along with their exponentially smoothed versions. The network's hidden layer consisted of five neurons with a hyperbolic tangent activation function. The output layer contained a single neuron and used a sigmoid activation function. The bend classifier also consisted of a linear threshold on the `asymmetry` feature. The classifier itself was exponentially smoothed over time, as described in the section above on multi-larva tracker feature extraction. Manual quantification of the single-larva tracker's bend classifier performance was based on 741 events from 10 larvae in 60 min of video data. This classifier has 97.6% precision and

**Table 2.** Fly crosses for larval experiments.

For strain information, see Key resources table.

| Figure | Designation | Female parent | Male parent |
|---|---|---|---|
| 2 | 72F11>CsChrimson | UAS-CsChrimson | 72F11-Gal4 |
| 2 | 69F06>CsChrimson | UAS-CsChrimson | 69F06-Gal4 |
| 2 | attP2 >CsChrimson | UAS-CsChrimson | attP2 |
| 2 | 72F11>dTrpA1 | UAS-dTrpA1 | 72F11-Gal4 |
| 2 | 69F06>dTrpA1 | UAS-dTrpA1 | 69F06-Gal4 |
| 2 | attP2 >dTrpA1 | UAS-dTrpA1 | attP2 |
| 2 sf1 | assorted genotypes | – | – |
| 3 | Or42b>CsChrimson, 72F11>dTrpA1 | Or42b>CsChrimson | 72F11>dTrpA1 |
| 3 | Or42b>CsChrimson, dTrpA1 | Or42b>CsChrimson | dTrpA1 |
| 4 | Ddc>CsChrimson | UAS-CsChrimson | Ddc-Gal4 |
| 4 | Ddc >CsChrimson, tsh >Gal80 | UAS-CsChrimson; tsh-LexA, LexAop-Gal80 | Ddc-Gal4 |
| 4 | 58E02>CsChrimson | UAS-CsChrimson | 58E02-Gal4 |
| 4 sf3 | Ddc >CsChrimson | UAS-CsChrimson | Ddc-Gal4 |
| 4 sf3 | CsChrimson control | UAS-CsChrimson | GMR-GAl4-attP2 |
| 5 | TH >CsChrimson | UAS-CsChrimson | TH-Gal4 |
| 5 | Tph >CsChrimson | UAS-CsChrimson | Tph-Gal4 |
| 5 | Tph >CsChrimson, tsh >Gal80 | UAS-CsChrimson; tsh-LexA, LexAop-Gal80 | Tph-Gal4 |

100% recall, with 99.2% accuracy of left and right detection during true-positive bends. The single-larva tracker also employed a ball classifier like that defined for the multi-larva tracker.

A user-defined Python software protocol operated on the behaviour detection output and specified closed-loop stimulus delivery instructions within the ROS framework. On this system, there existed a delay of up to 100ms between behaviour detection and closed-loop optogenetic stimulation.

## Larval rearing and handling

Fly stocks were maintained in vials filled with standard molasses food. In preparation for experiments and immunohistochemistry, adults (crosses listed in *Table 2*) were placed in collection cages with petri dishes containing molasses food and additional dry yeast to increase egg laying. For proof-of-principle and operant conditioning experiments, flies were allowed to lay eggs overnight for approximately 12–18 hr at 25°C. Larvae were reared at 25°C and experiments were performed 72–96 hr after egg laying. For temporal dynamics of larval heating and classical conditioning experiments, flies were allowed to lay eggs during daytime for approximately 7 hr at 25°C. Larvae were reared at 18°C and experiments were performed approximately seven days after egg laying. For immunohistochemistry, eggs were collected during daytime for approximately 4 hr. Dissections were performed 118–122 hr after egg laying. During all larval rearing, humidity was provided by placing wet paper towels placed beside the food plates. Specifically for optogenetics experiments, larvae were reared in the dark on molasses food supplemented with all-*trans*-retinal.

All larval handling and experiments were performed in the dark to avoid unintended optogenetic stimulation. For each experiment run performed on the high-throughput tracker, we extracted larvae from their food plate and washed them in water. Using a brush, we placed multiple larvae in the centre of the agarose plate. We then placed the agarose plate inside the tracker on top of the backlight and shut the tracker door tightly. For experiments on the low-throughput, single-larva tracker, we extracted larvae from their food plate using a 15% sucrose solution. For a given larva, time outside the food plate did not exceed 30 min prior to the experiment starting. We used a brush to extract a single larva from the sucrose solution and rinse it in water to remove residual sucrose. We then placed the single larva in the centre of the agarose plate and shut the tracker door tightly. Prior to starting all experiment runs, we gave larvae a short period of time to acclimate to their new environment and, in the case of high-throughput experiments, disperse so they were not touching each other.

## Verification of optogenetic and thermogenetic stimulation efficiency

### Experiment procedures

We assessed the multi-larva tracker's optogenetic and thermogenetic stimulation efficiency through open-loop proof-of-principle experiments. The 1 min experiment protocol began with a 15 s initialisation period in which larvae acclimated to the agarose plate and behaviour classifiers stabilised. In three subsequent 15 s stimulation rounds, larvae received 5 s of open-loop stimulation followed by 10 s without stimulation (*Figure 2C*). Optogenetics were performed with a combined red light intensity of 285 µW/cm² from two DMDs as described above. Thermogenetics were performed with the 1490 nm laser at 40% of its maximum available 5.26 W intensity.

### Data analysis

Analysis of these open-loop optogenetic and thermogenetic experiment data (*Figure 2D*, *Figure 2E*) was conducted using custom MATLAB software. After equally splitting each 60 s experiment into either 0.5 s or 5 s time bins, we retained objects for analysis that fulfilled strict criteria: (i) for all bins following the 15 s initialisation period, the object's initial detection must have occurred at least 15s prior to the start of the bin; (ii) the object must have been detected in every frame of the bin (i. e. it retained its identity and did not collide with another object); and (iii) the mean of the smoothed centroid velocity across the object's detection period in the bin was at least 0.5 mm/s. For each larva, the criterion for rolling, crawling, or bending was detection of the corresponding behaviour at least once during a given bin, irrespective of the behaviour's duration. The number of larvae performing each behaviour in a given bin was then divided by the total number of larvae in the bin to give the fractions displayed in *Figure 2D* and *Figure 2E*.

## Temporal dynamics of larval heating

### Experiment procedures and data analysis

To quantify the time course of larval heating with our setup (*Figure 2—figure supplement 1*), we performed additional open-loop stimulation experiments with the 1470 nm laser. The experiment protocol began with a 30 s initialisation period without heat stimulation. Then the IR laser was turned on at 50% intensity for 20 s. During this stimulation period, the heat camera was used to maintain an intended temperature of 30°Cat each larval location, as described above. Ten runs of this experiment were performed, with four larvae placed on the agarose plate per run. To analyse these data, recorded temperatures at each larval location for every point in time were collected.

## High-throughput classical conditioning

### Experiment procedures

Eight larvae were trained during each run of the classical conditioning experiment, with 18 runs for experimental larvae and 14 runs for control larvae. Training began after an initial 30 s initialisation period. Forward-paired training consisted of eight replicate training rounds, yielding a total training period lasting 840 s. Each round began with 20 s of red light illumination (via two 613nm DMDs with combined intensity of 550 µW/cm$^2$, as described above), followed by 5 s without stimulation and then 20 s of IR illumination (via 1470 nm laser). This IR illumination caused larval heating up to 27.5°C, which was maintained using the closed-loop temperature control system described above. Each round ended with 60 s without stimulation. Backward-paired training was identical to forward-paired with the exception that IR illumination occurred first, followed by red light illumination. 60 s without stimulation followed the training period. In the subsequent testing period, larvae were illuminated for 20 s with the same wavelength and intensity of red light as during training. The testing period involve three rounds of this illumination, with 45 s of no stimulation between each replicate (*Figure 3A*).

### Data analysis

For this analysis, we divided each testing round into three time windows: PRE (10–0 s before light onset), ON (0–20 s after light onset), and OFF (0–20 s after light offset). Only larvae that were tracked for all three ON and OFF windows during the testing period were analysed. To account for violated normality and sphericity assumptions, we calculated the difference in percentage bending values between the ON and OFF windows (i. e. ON-OFF) for each individual larva. These percentage bending differences were then compared between training protocols and genotypes to study the effects of training on larval bending behavior during the testing period. Because percentage bending differences were not normally distributed, we used two-sided Mann–Whitney *U* tests to assess how bending behaviour differed between training protocols and genotypes.

## High-throughput operant conditioning

### Experiment procedures

Each run of the operant conditioning experiment protocol (*Figure 4B*) consisted of 10–12 larvae on the plate. The protocol began and ended with a 1 min test period without optogenetic stimulation. Between these test periods were four, 3 min training rounds during which larvae received red light stimulation of 285 µW/cm$^2$ (combined from two DMDs as described above) for the entire duration of the detected bend. Which side received stimulation was randomised across trials such that approximately 50% of larvae were trained to develop a right bend preference and 50%a left bend preference. No stimulus was triggered when the larva was bending right or when its body was straight. The test periods were each separated by 3 min periods without stimulation. After the first minute of this period, a brush was used to gently move all larvae back to the centre of the plate and larvae were given time to recover before the beginning of the next training round. This recentring addresses problems encountered when performing prolonged experiments with freely behaving larvae on a small agarose plate. The longer larvae are left undisturbed, the more likely they are to touch the plate's edge, causing tracking disruption and temporary loss of valid objects. This shrinks sample size and reduces training efficiency by decreasing the proportion of animals receiving the stimulus.

Control experiments were designed so that valid objects received optogenetic stimulation uncorrelated with behaviour. These control experiments were split into 60 s time bins, during which each

valid object was randomly assigned a stimulus train from this same time bin, pulled from a prior experiment where stimulation correlated with behaviour.

## Data analysis

Data analysis was conducted using custom MATLAB software. After equally splitting each experiment into 60 s time bins, we retained objects for analysis that fulfilled strict criteria: (i) the object must have been detected in every frame of the bin (i.e. it retained its identity and did not collide with another object); (ii) the object's initial detection must have occurred at least 20 s prior to the start of the bin; (iii) at no point during the bin did the smoothed velocity of the larval centroid exceed 1.5 mm/s; and (iv) the mean of the smoothed centroid velocity across the object's detection period in the bin was at least 0.5 mm/s.

To analyse operant conditioning of bend direction preference, it was necessary to further smooth the raw time series of left and right bends post-acquisition: two bends to the same side separated by less than 200ms were combined into a single long bend, and short bends of less than 200ms were removed from analysis. We then counted, for each larva, the numbers of left and right bends initiated within each 60 s time bin. This was defined as the bend rate towards the respective direction. Within each bin, the difference in bend rate was defined, for each larva, as the number of bends towards the side paired with the optogenetic stimulus minus the number of bends towards the unstimulated side. We pooled together all larval data within each bin because bends to the left and right were each paired with the optogenetic stimulus for approximately half of the larvae. For the control condition in which larvae received random stimulation during 50% of bends regardless of direction, bend rates were calculated to the left and right and the difference in bend rate was calculated between left and right. Bend rates to either side were compared to each other using a two-sided Wilcoxon signed-rank test. This statistical choice was driven by the known pairing of these spatial observations for individual larvae and the non-normality exhibited across distributions of differences between these paired bend rates. Similar reasoning guided the usage of a two-sided Wilcoxon signed-rank test to compare the difference in bend rate to 0. The behaviour characteristics of experimental animals were compared to each control group using a two-sided Mann-Whitney $U$ test.

## Low-throughput operant conditioning

### Experiment procedures

Each run of the operant conditioning experiment protocol (*Figure 4—figure supplement 3*) consisted of one larva on the plate. The protocol began and ended with a 1 min test period without optogenetic stimulation. Between these test periods were two, 3 min training rounds during which larvae received red light stimulation of 385 µW/cm$^2$ for the entire duration of the detected bend. Which side received stimulation was randomised across trials such that approximately 50% of larvae were trained to develop a right bend preference and 50%a left bend preference. No stimulus was triggered when the larva was bending right or when its body was straight. The two test periods were separated by a 3-min period without stimulation.

### Data analysis

Data analysis was conducted using custom MATLAB software. Each experiment was equally split into 60 s time bins. We then counted, for each larva, the numbers of left and right bends initiated within each 60 s bin. This was defined as the bend rate towards the respective direction. Within each bin, the difference in bend rate was defined, for each larva, as the number of bends towards the side paired with the optogenetic stimulus minus the number of bends towards the unstimulated side. We pooled together all larval data within each bin because bends to the left and right were each paired with the optogenetic stimulus for approximately half of the larvae.

## Immunohistochemistry and confocal imaging

All dissections, immunohistochemical stainings, and confocal imaging were done using a procedure adapted from *Jenett et al., 2012* and *Li et al., 2014*. Larval CNSs were dissected in cold 1x phosphate buffer saline (PBS, Corning Cellgro, #21–040) and transferred to tubes filled with cold 4% paraformaldehyde (Electron Microscopy Sciences, #15,713S) in 1x PBS. Tubes were incubated for 1 hr at room temperature. The tissue was then washed four times in 1x PBS with 1% Triton X-100 (#X100,

Sigma Aldrich St. Louis, Missouri) (PBT) and incubated in 1:20 donkey serum (#017-000-121, Jackson Immuno Research, West Grove, Pennsylvania) in PBT for 2 hr at room temperature.

The tissue was then incubated in the primary antibody solution, first for 4 hr at room temperature and then for two nights at 4°C. This solution contained mouse anti-Neuroglian (1:50, #BP104 anti-Neuroglian, Developmental Studies Hybridoma Bank, Iowa City, Iowa), rabbit anti-GFP (1:500, #A11122, Life Technologies, Waltham, Massachusetts) and rat anti-N-Cadherin (1:50, #DN-Ex #8, Developmental Studies Hybridoma Bank, Iowa City, Iowa) in PBT. This solution was then removed and the tissue washed four times in PBT. The tissue was then incubated in the secondary antibody solution, first for 4 hr at room temperature and then for two nights at 4°C. This solution contained Alexa Fluor 568 donkey anti-mouse (1:500, #A10037, Invitrogen, Waltham, Massachusetts), FITC donkey anti-rabbit (1:500, #711-095-152, Jackson Immuno Research West Grove, Pennsylvania) and Alexa Fluor 647 donkey anti-rat (1:500, #712-605-153, Jackson Immuno Research West Grove, Pennsylvania) in PBT. After removal of the secondary solution, the tissue was washed in PBT four times and mounted on a coverslip coated with poly-L-lysine (#P1524-25MG, Sigma Aldrich, St. Louis, Missouri).

The coverslip with the CNSs was dehydrated by moving it through a series of jars containing ethanol at increasing concentrations (30%, 50%, 75%, 95%, 100%, 100%, 100%) for 10 min each. The tissue was then cleared by soaking the coverslip with xylene (#X5-500, Fisher Scientific, Waltham, Massachusetts) three times for 5 min each. Finally, the coverslips were mounted in dibutyl phthalate in xylene (DPX, #13512, Electron Microscopy Sciences, Hatfield, Pennsylvania) with the tissue facing down on a microscope slide with spacers. The DPX was allowed to dry for at least two nights prior to confocal imaging with an LSM 710 microscope (Zeiss).

Details on the confocal imaging settings are provided in the respective figure captions. Confocal images were analysed using Fiji (ImageJ). Neurons were counted by specifying regions of interest around the cell bodies using raw image stacks.

## Acknowledgements

We thank Dr. Chris McRaven for design and technical assistance with the custom-built 1490nm thermogenetic laser light source; Dr. Peter Polidoro for writing the ROS software framework on the single-larva tracker; Howard Hughes Medical Institute (HHMI) Janelia FlyCore and FlyLight teams for assistance with fly crosses, fly food, and confocal imaging; Gates Cambridge Trust, Cambridge Trust, HHMI Janelia Visiting Scientist Program, University of Cambridge Trinity College, HHMI Janelia, European Research Council, Wellcome Trust, and Medical Research Council for funding.

## Additional information

### Competing interests

Lalanti Venkatasubramanian: Marta Zlatic: The other authors declare that no competing interests exist.

### Funding

| Funder | Grant reference number | Author |
| --- | --- | --- |
| Gates Cambridge Trust | | Kristina T Klein |
| Cambridge Commonwealth, European & International Trust | C T Taylor Cambridge International Scholarship | Elise C Croteau-Chonka |
| Howard Hughes Medical Institute Janelia Research Campus Visitor Scientist Program | | Marta Zlatic Jean-Baptiste Masson |
| Trinity College, University of Cambridge | | Elise C Croteau-Chonka |
| Howard Hughes Medical Institute Janelia Research Campus | | Marta Zlatic Michael Winding |

| Funder | Grant reference number | Author |
|---|---|---|
| European Research Council | LeaRNN - 819650 | Marta Zlatic |
| Wellcome Trust | 205050/B/16/Z | Marta Zlatic<br>Lalanti Venkatasubramanian<br>Elise C Croteau-Chonka<br>Michael Winding |
| Medical Research Council | MC_UP_1201/20 | Marta Zlatic<br>Michael S Clayton<br>Benjamin MW Jones<br>Samuel N Harris |
| Human Frontier Science Program | LT00600/2020-L | Lalanti Venkatasubramanian |

The funders had no role in study design, data collection and interpretation, or the decision to submit the work for publication. For the purpose of Open Access, the authors have applied a CC BY public copyright license to any Author Accepted Manuscript version arising from this submission.

## Author contributions

Elise C Croteau-Chonka, Data curation, Software, Formal analysis, Validation, Investigation, Visualization, Methodology, Writing – original draft, Writing – review and editing; Michael S Clayton, Conceptualization, Resources, Data curation, Software, Formal analysis, Supervision, Validation, Investigation, Visualization, Methodology, Writing – original draft, Project administration, Writing – review and editing; Lalanti Venkatasubramanian, Conceptualization, Resources, Supervision, Validation, Investigation, Methodology, Project administration; Samuel N Harris, Conceptualization, Methodology; Benjamin MW Jones, Conceptualization, Investigation; Lakshmi Narayan, Michael Winding, Jean-Baptiste Masson, Resources, Software, Methodology, Writing – review and editing; Marta Zlatic, Conceptualization, Supervision, Funding acquisition, Project administration, Writing – review and editing; Kristina T Klein, Conceptualization, Data curation, Software, Formal analysis, Validation, Investigation, Visualization, Methodology, Writing – original draft, Writing – review and editing

## Author ORCIDs

Elise C Croteau-Chonka http://orcid.org/0000-0001-5116-3772
Michael S Clayton http://orcid.org/0000-0003-1152-9199
Lalanti Venkatasubramanian http://orcid.org/0000-0002-9280-8335
Marta Zlatic http://orcid.org/0000-0002-3149-2250
Kristina T Klein http://orcid.org/0000-0002-8772-3628

## Decision letter and Author response

Decision letter https://doi.org/10.7554/eLife.70015.sa1
Author response https://doi.org/10.7554/eLife.70015.sa2

---

# Additional files

## Supplementary files

• Transparent reporting form

## Data availability

All data used to generate figures 2-5, as well as all figure supplements, are now submitted as source data files. We also now submit CAD drawings for the multi-larva tracker.

---

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
