## [Editor Report]

Since classic studies by Pavlov and Skinner about learning and memory in laboratory settings, the field has sought solid ways to use observable behaviors to illuminate how positive and negative reinforcement modulates stimulus-evoked behaviors. This valuable study by Croteau-Chonka et al., represents the latest modernization of such technology, judiciously targeting the highly quantifiable, real-time motor behaviors of the *Drosophila* larva to create a rigorous new paradigm for classical and operant conditioning. The small nervous system of the larva has few parallels for tractability in dissecting various neural mechanisms. What has been lacking, until now, is the technology needed to tackle the neural mechanisms of learning and memory in the maggot with rigorous, high-throughput, and real-time behavioral observations.

---

## [Decision Letter]

**Decision letter after peer review:**

Thank you for submitting your article "Serotonergic neurons mediate operant conditioning in *Drosophila* larvae" for consideration by *eLife*. Your article has been reviewed by 3 peer reviewers, and the evaluation has been overseen by a Reviewing Editor and K VijayRaghavan as the Senior Editor. The reviewers have opted to remain anonymous.

Essential revisions:

1) Because the optogenetically induced operant learning has a modest effect, the Reviewers would like to see replication of operant conditioning using the thermogenetic paradigm (shown to be effective in Figure 2).

2) More rigorous analyses of the phenotype (e.g., showing pre and postbending behaviors, see Reviewer #1… additional statistical tests, see Reviewer #2… additional controls, see Reviewer #3).

3) All reviewers were concerned about the artificial nature of the US. If operant conditioning can be evoked with nociceptive pain, that would be great. If not, a more careful discussion of the limitations of using artificial stimuli in the interpretation and discussion would suffice.

*Reviewer #1 (Recommendations for the authors):*

Below I will list the criticism summarized in the previous section

1) To validate the set up the authors induce rolling behavior by thermo- or optogenetically activating two sets of previously described neurons in individual larva. Both approaches show convincing induction of the behavior per se. However, there seems to be an interaction of the different tools used (thermo and opto-genetic) and the targeted neurons: the authors observe different dynamics of the behavior across the three stimulation cycles depending on stimulation method and labeled neurons. These findings make it difficult to understand why the authors choose only the optogenetic activation to investigate operant conditioning. Thus, the authors should repeat key experiments using thermogenetic tools (see below).

2) The strength of the setup is that individual animals can be targeted. Though the presented data show that behavior can be reliably induced in stimulated animals, it lacks the information about the behavior of non-targeted larva during the stimulation. Thus, it would strengthen the work if the authors could show the behavior of the non-targeted larva during the time when targeted larva receive light or heat.

3) The origin of the relative difference between left and right bending in the paired group is not entirely clear. Thus, it will be important to strengthen the work by additional experiments investigating temporal relationship between the CS and US. Ideally the experiments should focus on understanding why the training results in a reduction of the unreinforced behavior.

4) The authors should consider that given the small effects, it has to be ensured that the observed differences originate from training and are not mere pre-training biases. Though they show the pretraining results for one of the experiments (Figure 3b, the trained group), the pretraining bending is very relevant for each of the operant learning experiments. In fact, training induced effects should not only be measured by looking at the left vs right bending in the final test but as a change between pre versus post or between a trained and a mock control group. This is done for one group (Ddc-GAL4) in Figure 3b but will be mandatory for all operant learning experiments.

5) It would improve the accessibility of the learning induced change of behavior if the authors could show the pre vs post training results for each run (10-12 larva in a plate). Further, they should plot the numbers of reinforced behaviors in each of the training protocols and relate it to the test performance.

6) The presented data clearly suggests a decrease of the unstimulated bending rather than a change in the reinforced behavior. Though the authors mention it, they do not explain or discuss it. It will be very important for the logic of the manuscript that the authors explain this phenomenon and how it relates to operant conditioning (see point 3).

7) Though the manuscript discusses most of the data carefully, in my view the authors miss an important issue: it remains to be shown if fly larva are capable of operant learning using external reward or punishment. The presented evidence is based on artificial activation of neurons, which arguably is a hint but not a prove that operant conditioning is withing the repertoire of a fly larva, an issue the authors should mention and discuss.

*Reviewer #2 (Recommendations for the authors):*

In addition to the points raised in the public review:

– In figure 2, I was surprised to see that the responses elicited by the same Gal4 are so different when stimulated optogenetically or thermogenetically for both lines. Can the authors try to give an explanation?

– I am intrigued by what would be the US under natural conditions. In the discussion the authors say that they did not manage to condition the larvae using sugar sensing neurons, could it be that in *Drosophila* larvae operant conditioning only works for sensory signals imputing directly to the VNC? Have they tried conditioning the larvae by associating one side bending with optogenetic activation of noxious stimuli (without eliciting rolling)? It might be good to expand in the discussion on the expected sensory nature of US enabling operant conditioning.

*Reviewer #3 (Recommendations for the authors):*

The introduction is an excellent description of the central question – can *Drosophila* larvae undergo operant conditioning and if so, what neural circuits enable it? The background for the types of learning, the known and possible circuit mechanisms, and the advantages of the model system, are beautifully described (easily Review-caliber) allowing diverse readers to engage with the paper.

To conduct operant learning experiments, new hardware and software were required. Tracking multiple larva, identifying their behaviors, and applying rewarding or punishing stimuli through optogenetic activation of neurons – all in real-time – is a tour de force. Again, the descriptions of the key hardware innovation (the field-programable gate arrays) and the advances in machine learning enabling rapid behavior detection are very nicely done.

Line 210: Good control showing that the IR laser activates Trp but not pain-induced rolling. (The thermogenetic activation isn't really key to the later experiments, though…)

Recommended Revisions:

The experiments are solid and rigorously done. Most of the issues can be resolved with text clarification or additional discussion.

The best interpretation of the behaviors induced by expression in many dopaminergic and serotoninergic neurons is unclear: different neurons can have local functions – sometimes conflicting ones – so I am surprised that a very broad driver like DDC doesn't set up conflicting or non-specific reinforcement. I am also amazed that the timing works – a reward right after the bend could generate a different association than a potential punishment right before it. The interval between bends and the timing of the optogenetic reinforcement (assumed to be positive) relative to those bends should be discussed.

How many behavioral actions do you miss applying reward or punishment to? There are multiple larvae on the plate, and the bends must be detected and then the larva targeted – when you look at the videos later, what fraction of the bends that occur during the training periods were successfully reinforced?

How was the training regimen selected? What about a longer, continuous period – or shorter breaks? Was there a minimum number of reinforced bends that had to occur to include a larva in the subsequent evaluation? Was the possibility that the un-reinforced epochs in the training regime result in forgetting (or learning that action does not reliably produce reward) considered?

Important missing control: There have been reports that the UAS-Chrimson can have some basal/leak expression. What does the attP2>Chrimson + retinal control look like in the biasing for bend direction experiments (Figure 3B)? Even a low level of expression could serve as a reward and so this control should be tested. (The uncorrelated light control is also good, but does not rule out contribution from activation of non-serotonergic neurons.)

line 289 (and line 395): The use of necessity and sufficiency is a little confusing here. These are all ectopic manipulations. Optogenetic activation of all dopaminergic and serotoneric neurons (expressing DDC-GAL4, UAS-Chrimson) is capable of inducing increased bending. Activation of dopaminergic and serotonergic neurons in the brain alone (DDC-GAL4, UAS-Chrimson, tsh-LexA, LexOp-GAL80) does not. We don't know much about how these neurons contribute to normal operant conditioning, just that the ones in the VNC seem to be the critical ones for the optogenetic effect. It could be said that activation of serotonergic VNC neurons is sufficient to induce the optogenetic effect – but I would avoid any speculation of actual necessity. (line 375 described operant conditioning as "impaired" which is also confusing. The artificial operant conditions failed when VNC serotonergic neurons were not included. That is not the same thing.) line 413: optogenetic activation CAN serve, not DOES serve…. This all may be a difference of word use conventions among research studying learned vs. innate behaviors, but given the discussion about value of these terms (Gomez-Marin 2017, Yoshihara and Yoshihara 2018), it might be good to maximize clarity here.

Operant conditioning is measured as an increase in number of bends per minute or in the probability of bending toward the rewarded side. These are not directly comparable metrics to the performance index used in classical conditioning, but the operant effects, while statistically significant, seem very small (eg. Figure 4b vs. d). Is this an accurate observation and is there a useful contrast proposed?

I found the shifts between dopaminergic and serotonergic neurons, and between classical and operant conditioning experiments, a little challenging to follow. Please consider clarifying how the classical conditioning experiments contribute to the focus of this research that is suggested by the title – serotonergic neurons capable of operant conditioning. (Since it is clear early on that operant and classical conditioning are achieved by different neurons, it is not logical that the classical conditioning screen for subsets of dopaminergic or serotonergic neurons would turn up any useful candidates for dissecting operant circuits.)

The Tsh-LexA, lexOp-GAL80 combination should reference Simpson 2016 J Neurogenetics; only the Tsh-GAL80 was published in Clyne and Miesenbock. The Tsh-LexA, lexOp-flp, UAS>stop>Chrimson combination, the otd-Flp (Asahina et al., 2014), or the Trh-DBD+Tsh-AD (Albin et al., 2015) present possible options for positive intersection with serotonergic neurons in the VNC or the brain.

Suggestions for future experiments (not required):

Line 256: These larvae don't show innate bend direction preference. Out of curiosity, would it be possible to try this on DeBivort's turn preference-prone (handed) animals? Is it easier to train toward an existing bias or totally distinct? would this help identify shared/distinct circuit contributions?

What happens to the larval bend rates and direction preference if you silence serotonergic neurons? Are they learning anything by their normal crawling actions? Is there a paradigm where operant learning might occur in a more natural or beneficial context? The experiments described here show that larvae are capable of it in a highly artificial setting – which is certainly cool – but provide no clues about what larva might be using this capacity for in their normal lives…

What about removing or silencing the MBs altogether and seeing if you can still optogenetically induce operant conditioning? That would be a pretty nice way to show that not all roads lead to MBs!

---

## [Author Response]

Essential revisions:1) Because the optogenetically induced operant learning has a modest effect, the Reviewers would like to see replication of operant conditioning using the thermogenetic paradigm (shown to be effective in Figure 2).

In order to address this comment, we first characterised in more detail the temporal profile of heating with the IR laser beam. To verify the speed with which the IR beam heated up individual animals we installed a heat camera on the setup. Using the heat camera, we found that the time to heat up the animal to the desired 30^o^C is 4s. The heat camera performed closed-loop adjustments of laser intensity to maintain the desired temperature at each larval location.

While larvae were heated to the right temperature relatively quickly and with predictable timing relative to IR light trigger (within 4s), the delay between trigger and thermogenetic stimulation was much slower than for optogenetic stimulation (which is virtually instantaneous). Since the average duration of a larval bend is only a third of that time (mean=1.35s, sd=1.67s, n=4622 bends), thermogenetic activation is too slow to be used in closed-loop during operant conditioning. By the time the fictive reward would be activated, the animal could be performing a completely different action, which could interfere with the formation of an association with the preceding action.

A second important methodological consideration was the risk of establishing conflicting valence signals. We wanted to avoid mixing punishment (via IR-induced tissue heating) with reward (via IR-induced activation of Ddc neurons) during training. With mild heat more aversive (induces more bending in control animals) to larvae than visible red light (compare bending in *attp2* control larvae in Figure 2D and 2E), we favoured optogenetics over thermogenetics for the operant conditioning paradigm.

We therefore decided against trying to replicate the operant conditioning experiments using thermogenetic stimulation.

We have added new Figure 2 —figure supplement 1 and the following new sections in the Results explaining these points:

“The methodological choice of optogenetics was informed, in part, by a deeper investigation of larval heating dynamics as they relate to average bend duration. We determined that our IR stimulation hardware takes approximately 4s to heat larval tissue to the nearly 30^o^C required for dTrpA1 channel activation (Figure 2 —figure supplement 1; see also Materials and methods). Knowing that the average duration of a larval bend is only a third of that time (mean=1.35s, sd=1.67s, n=4622 bends), we concluded that closed-loop heat stimulation would activate neurons of interest only after a noticeable delay relative to behaviour detection, with possibly other behaviours occurring during the delay period. Such a task in which different behaviours are occurring prior to reinforcement could be very difficult to learn. Quicker heating was achievable with increased laser intensity, but such an approach risked overshooting the desired temperature and damaging larval tissue. Any safe thermogenetic approach would therefore be too slow to temporally align the US induced via larval heating with a specific larval action. A second important methodological consideration was the risk of establishing conflicting valence signals. We wanted to avoid mixing punishment (via IR-induced tissue heating) with reward (via IR-induced activation of Ddc neurons) during training. With mild heat more aversive to larvae than visible red light (compare bedning in control larvae in Figure 2D and 2E), we favoured optogenetics over thermogenetics for our operant conditioning paradigm.”

However, we do agree that replicating the operant learning experiments in a different way is important.

To address this point and the point about needing the effector control (from Reviewer 3), we therefore reproduced the operant learning experiment using optogenetic activation, but on a different, low-throughput, single-animal tracker (Schulze et al., 2015).

As before, we found that pairing optogenetic activation of dopaminergic and serotonergic neurons (Ddc-GAL>UAS-CsChrimson larvae) with a specific bend direction resulted in a significant increase in bends/min towards the stimulated side relative to the unstimulated side, after training, but not before. This indicates operant learning has occurred (Figure 4—figure supplement 3d).

In contrast, there was no significant increase in bends/min towards the stimulated side relative to the unstimulated side, neither after, nor before training in the UAS-CsChrimson effector control larvae (Figure 4—figure supplement 3e). These results show that putative leaky CsChrimson expression in some unknown neurons is not sufficient to induce operant learning.

Furthermore, after training, there was a significant difference in Δ bend/min between Ddc-GAL4>UAS-CsChrimson and the UAS-CsChrimson larvae (Figure 4—figure supplement 3f). This confirms that optogenetic activation of Ddc serotonergic and dopaminergic neurons can induce operant learning.

We have added new Figure 4 —figure supplement 3 with these results and a new section in the Results:

“We also used a previously developed low-throughput single-larva closed-loop tracking system to test the reproducibility of this result on a different system (see Materials and methods for more details, Schulze et al., 2015). Fictive Ddc activation with this low-throughput system also yielded a significant bend direction preference to the previously stimulated (mean=5.77, sd=2.71, n=109) versus previously unstimulated (mean=4.73, sd=2.73, n=109) side (p=.0043), after training. These results contrast those of effector control larvae that had the UAS-CsChrimson transgene but not the Ddc-GAL4. The control larvae show no difference in bend rate to either side after training. Based on these control larvae, we concluded that potential basal expression of CsChrimson in neurons outside of the Ddc expression pattern is not causing operant learning (Figure 4 —figure supplement 3).”

Finally, we note that, while thermogenetic activation is not fast enough for closed-loop operant conditioning, it is well suited for open-loop experiments and for combining thermogenetic with optigenetic activation. Since the timing between triggering and heating is constant and predictable (4s) it is possible to design classical conditioning experiments with desired CS-US timing intervals.

Our prior version of the manuscript did not demonstrate the usefulness of our new IR thermogenetic stimulation module for classical conditioning paradigms. We wanted to address this point by combining both fictive thermogenetic and optogenetic activation to demonstrate a previously unknown form of classical conditioning in larvae.

Trace conditioning has been demonstrated in adult *Drosophila* (Galili et al., 2011) but whether or not larvae can associate a CS with a US when there is a significant temporal gap between CS offset and US onset was previously unknown. We therefore trained larvae using a trace conditioning paradigm in which we combined fictive odour (optogenetic activation of Or42b) with fictive noxious stimulus (thermogenetic activation of the nociceptive and mechanosensory Basin neurons) with a 9s gap between CS offset and US onset. In these experiments, we triggered IR light 5s after the offset of red light for optogenetic stimulation and it took a further 4s to reach the appropriate 30^o^C for thermogenetic activation of Basins.

In this way, we demonstrate for the first time that larvae are capable of trace conditioning (i.e. associating a CS with a US after a 9s temporal gap between the CS offset and US onset).

We have added a new Figure 3 and a new section entitled “Aversion to fictive Or42b develops after forward-paired trace conditioning” that demonstrate trace-conditioning using optogenetic odour and thermogenetic noxious stimulation.

“Having verified the efficacy of optogenetic and thermogenetic stimulation in our system, we first studied whether these methods could be used to train larvae in a previously unexplored classical conditioning task that requires precise temporal control of both CS and US. […] These results show that our tracker can be used to perform automated, high-throughput classical conditioning in *Drosophila* larvae. To the best of our knowledge, these results also provide the first evidence that larvae can perform classical conditioning with significant (9s) offset-to-onset gaps between stimuli (i.e. trace conditioning).”

2) More rigorous analyses of the phenotype (e.g., showing pre and postbending behaviors, see Reviewer #1… additional statistical tests, see Reviewer #2… additional controls, see Reviewer #3).

We have expanded Figures 4c-f and 5a-c to show bend/min towards the stimulated and unstimulated side and compute the difference between them and test whether this difference is significant (using two-sided Wilcoxon signed-rank test), both before and after training, for all genotypes and all conditions. We also compare (using a two-sided Mann-Whitney U test, with Bonferroni correction) the difference in bends/min towards the stimulated and unstimulated side (Δ bends/min), after training, between experimental and control groups (Figures 4h and 5d), as suggested by the Reviewers.

Pairing optogenetic activation of dopaminergic and serotonergic (Ddc-GAL>UAS-CsChrimson larvae, Figures 4c) or just serotonergic neurons (Tph-GAL4>UAS-CsChrimson, Figure 5a) with a specific bend direction, resulted in a significant increase in bends/min towards the stimulated side relative to the unstimulated side, after training, but not before. These results suggest that pairing a specific bend direction with the activation of serotonergic and dopaminergic neurons together, or serotonergic neurons alone, induces operant learning.

For all other genotypes there was no significant difference in bends/min to the stimulated, compared to the unstimulated side, neither before, nor after training. Thus, pairing the activation of only brain dopaminergic and sertotonergic (Ddc-GAL4>UAS-CsChrimson, teashirt-GAL80, Figure 4e), only MB dopaminergic neurons (58E02-GAL4>UAS-CsChrimson, Figures 4f), only dopaminergic neurons (TH-GAL4>CsChrimson, Figure 5b), or only brain serotonergic neurons (Tph-GAL4>UAS-CsChrimson, teashirt-GAL80, Figure 5c) with a specific bend direction did not result in a significant difference in bend direction between the stimulated and the unstimulated side, neither before, nor after training. These results indicate that none of these neuronal subsets alone were sufficient to induce operant learning. Since all of these genotypes contained the UAS-CsChrimson transgene, these results further suggest that any putative leaky basal CsChrimson expression in some unknown neurons is not sufficient to induce operant learning.

Similarly, the yoked control larvae (Ddc-GAL4>UAS-CsChrimson, Figure 4d) in which optogenetic activation of dopaminergic and serotonergic neurons was not paired with a specific bend direction did not result in a significant difference in bend direction between the stimulated and the unstimulated side, neither before, nor after training. This shows that simply optogenetic activation of dopaminergic and serotonergic neurons unpaired with bend does not bias bend direction.

We also compared Δ bends/min after training between the experimental and control genotypes conditions and found they were significantly different.

Thus, after training, there was a significant difference in Δ bend/min between

– the Ddc-GAL4>UAS-CsChrimson and the Ddc-GAL4>UAS-CsChrimson, teashirt-GAL80 (Figure4h).

– the Ddc-GAL4>UAS-CsChrimson and the 58E02-GAL4>UAS-CsChrimson (Figure 4h).

– the Ddc-GAL4>UAS-CsChrimson paired larvae and the yoked controls (Figure 4h).

– the Tph-GAL4>UAS-CsChrimson and the Tph-GAL4>UAS-CsChrimson, teashirt-GAL80 (Figure 5d).

These results are consistent with the idea that only the complete set of serotonergic neurons is sufficient to induce operant learning. In the absence of activation of, either all serotonergic, or just nerve cord serotonergic neurons, we did not observe operant learning.

Finally, we also added the additional control, together with demonstrating the reproducibility of operant learning on a different, low-throughput, single-larva tracker (Schulze et al., 2015). We have paired a specific bend direction with red light, either in the Ddc-GAL4>UAS-CsChrimson larvae, or in larvae in which UAS-CsChrimson was present, but Ddc-GAL4 was absent (Figure 4–figure supplement 3).

As before, we found that pairing optogenetic activation of dopaminergic and serotonergic neurons (Ddc-GAL>UAS-CsChrimson larvae) with a specific bend direction resulted in a significant increase in bends/min towards the stimulated, relative to the unstimulated side, after training, but not before, indicating operant learning has occurred (Figure 4–figure supplement 3d).

In contrast, there was no significant increase in bends/min towards the stimulated, relative to the unstimulated side, neither after, nor before training in the UAS-CsChrimson control larvae (Figure 4–figure supplement 3e). These results further support the idea that any putative leaky basal CsChrimson expression in some unknown neurons, is not sufficient to induce operant learning.

Furthermore, after training, there was a significant difference in Δ bend/min between Ddc-GAL4>UAS-CsChrimson and the UAS-CsChrimson larvae (Figure 4–figure supplement 3f). This confirms that optogenetic activation of Ddc serotonergic and dopaminergic neurons can induce operant learning.

These results and quantifications are shown in the new Figure 4–figure supplement 3.

3) All reviewers were concerned about the artificial nature of the US. If operant conditioning can be evoked with nociceptive pain, that would be great. If not, a more careful discussion of the limitations of using artificial stimuli in the interpretation and discussion would suffice.

To address this point, we attempted preliminary experiments, either with noxious heat (data not shown) or with optoegentic activation of MD class IV neurons (targeted by the ppk-Gal4 driver, preliminary data not shown ) during bends to one side to see whether these neurons could evoke aversive operant learning. We were unable to elicit an operant learning effect in these pilot experiments, as evidenced by no significant difference in bend rate between the previously stimulated and unstimulated sides, after training (preliminary data not shown ). We cannot exclude that trying a range of different stimulation and training conditions with nociceptive heat or with optogenetic activation of MD class IV neurons could give operant aversive learning in the future, but it was beyond the scope of this study to explore this.

We have therefore added new sentences in the Discussion stating the limitations of artificial stimuli:“It will also be important to determine whether conditioning with a fictive odour stimulus changes larval responses to natural odours. For example, larvae showing reduced attraction to Or42b activation may show reduced attraction to natural odours that activate Or42b neurons (e. g. ethyl acetate)”

“There are limitations to interpreting behaviour following fictive neuronal activation. It will therefore be important to replicate the new paradigms developed with this rig using lower-throughput methods with natural stimuli. For example, one might modify our multi-larva tracker to accommodate closed-loop presentation of real, innately appetitive gustatory stimuli by using closed-loop control of microfluidic devices”

Reviewer #1 (Recommendations for the authors):Below I will list the criticism summarized in the previous section1) To validate the set up the authors induce rolling behavior by thermo- or optogenetically activating two sets of previously described neurons in individual larva. Both approaches show convincing induction of the behavior per se. However, there seems to be an interaction of the different tools used (thermo and opto-genetic) and the targeted neurons: the authors observe different dynamics of the behavior across the three stimulation cycles depending on stimulation method and labeled neurons. These findings make it difficult to understand why the authors choose only the optogenetic activation to investigate operant conditioning. Thus, the authors should repeat key experiments using thermogenetic tools (see below).

In order to address this comment, we first characterised in more detail the temporal profile of heating with the IR laser beam. To verify the speed with which the IR beam heated up individual animals, we installed a heat camera on the setup. Using the camera, we found that the time to heat the animal to the desired 30^o^C is 4s. The heat camera performed closed-loop adjustments of laser intensity to maintain the desired temperature at each larva.

While larvae were heated to 30^o^C relatively quickly and with predictable timing relative to IR trigger (within 4s), the delay between trigger and thermogenetic stimulation (which occurs once 30^o^C is reached) was greater than for optogenetic stimulation (which is virtually instantaneous). Since the average duration of a larval bend is only a third of that time (mean=1.35s, sd=1.67s, n=4622 bends), thermogenetic activation is too slow to be used in closed-loop during operant conditioning. By the time the reward neurons would be activated, the animal could be performing a completely different action, which could interfere with the formation of an association with the preceding action.

A second important methodological consideration was the risk of establishing conflicting valence signals. We wanted to avoid mixing punishment (via IR-induced tissue heating) with reward (via IR-induced activation of Ddc neurons) during training. With mild heat more aversive (induces more bending) to larvae than visible red light (compare bending in *attp2* control larvae in Figure 2D and 2E), we favoured optogenetics over thermogenetics for appetitive operant conditioning.

We therefore decided not to replicate the operant conditioning experiments using thermogenetics.

We have added new Figure 2 —figure supplement 1 and the following new sections in the Results explaining these points:

“The methodological choice of optogenetics was informed, in part, by a deeper investigation of larval heating dynamics as they relate to average bend duration. […] With mild heat more aversive to larvae than visible red light (compare bedning in control larvae in Figure 2D and 2E), we favoured optogenetics over thermogenetics for our operant conditioning paradigm.”

However, we agree that replicating the operant learning experiments in a different way is important. To address this point, we reproduced the operant learning experiment using optogenetic activation on a different, low-throughput, single-animal tracker (Schulze et al., 2015). Using this system we paired specific bend direction with red light, either in *Ddc-GAL4, UAS-CsChrimson* larvae, or in the effector control larvae that had the *UAS-CsChrimson* transgene but not the *Ddc-GAL4*. Pairing of Ddc activation with a specific bend direction on this low-throughput system also yielded a significant bend direction preference to the previously stimulated relative to the previously unstimulated side (p=.0043), after training. The effector control larvae showed no difference in bend rate to either side, after training. Based on these control larvae, we concluded that any putative basal expression of CsChrimson in neurons outside of the Ddc expression pattern cannot induce operant learning.

We have added new Figure 4 —figure supplement 3 with these results and a new paragraph in the Results section entitled “Operant conditioning of larval bend direction”:

“We also used a previously developed, low-throughput, single-larva, closed-loop tracking system to test the reproducibility of this result on a different system (see Materials and methods for more details, Schulze et al., 2015). Fictive Ddc activation with this system also yielded a significant bend direction preference to the previously stimulated (mean=5.77, sd=2.71, n=109) versus previously unstimulated (mean=4.73, sd=2.73, n=109) side (p=.0043), after training. These results contrast those of effector control larvae that had the UAS-CsChrimson transgene but not the Ddc-GAL4. The control larvae showed no difference in bend rate to either side after training. Based on these control larvae, we concluded that any potential leaky expression of CsChrimson in neurons outside of the Ddc expression pattern is not causing operant learning (Figure 4 —figure supplement 3).”

Finally, we note that, while thermogenetic activation is not fast enough for closed-loop operant conditioning, it is well suited for open-loop applications in combination with optigenetic activation. Since the timing between triggering and heating is predictable (4s), it is possible to design classical conditioning experiments with desired CS-US timing intervals.

Our prior version of the manuscript did not demonstrate the usefulness of our new IR thermogenetic stimulation module for classical conditioning paradigms. We, therefore, wanted to address this point by combining both fictive thermogenetic and optogenetic activation to demonstrate a previously unknown form of classical conditioning in larvae.

Trace conditioning has been demonstrated in adult *Drosophila* (Galili et al., 2011), but whether or not larvae can associate a CS with a US when there is a significant temporal gap between CS offset and US onset, was previously unknown. We therefore trained larvae using a trace conditioning paradigm in which we combined fictive odour (optogenetic activation of Or42b) with a fictive noxious stimulus (thermogenetic activation of the nociceptive and mechanosensory Basin neurons) with a 9s gap between CS offset and US onset. In these experiments, we triggered IR light 5s after the offset of red light for optogenetic stimulation and it took a further 4s to reach the appropriate 30^o^C for thermogenetic activation of Basins.

We found that larvae were capable of trace conditioning (associating a CS with a US after a 9s temporal gap between CS offset and US onset).

We have added the new Figure 3 and the new section entitled “Aversion to fictive Or42b develops after forward-paired trace conditioning” that demonstrate trace-conditioning using optogenetic odour and thermogenetic noxious stimulation.

“Having verified the efficacy of optogenetic and thermogenetic stimulation in our system, we first studied whether these methods could be used to train larvae in a previously unexplored classical conditioning task that requires precise temporal control of both CS and US. […] These results show that our tracker can be used to perform automated, high-throughput classical conditioning in *Drosophila* larvae. To the best of our knowledge, these results also provide the first evidence that larvae can perform classical conditioning with significant (9s) offset-to-onset gaps between stimuli (i.e. trace conditioning).”

2) The strength of the setup is that individual animals can be targeted. Though the presented data show that behavior can be reliably induced in stimulated animals, it lacks the information about the behavior of non-targeted larva during the stimulation. Thus, it would strengthen the work if the authors could show the behavior of the non-targeted larva during the time when targeted larva receive light or heat.

In the proof-of-principle experiments, all larvae receive light or heat during the 5 sec of open-loop stimulation. In other words, there do not exist non-targeted larvae at the same time as targeted larvae. However, we have expanded the time frame for which we show data in the Figure 2 to include times when the stimulus is *off*. This highlights baseline larval behaviour in the absence of fictive stimuli. We now show rolling, bending, and crawling before, during and after stimulation.

3) The origin of the relative difference between left and right bending in the paired group is not entirely clear. Thus, it will be important to strengthen the work by additional experiments investigating temporal relationship between the CS and US. Ideally the experiments should focus on understanding why the training results in a reduction of the unreinforced behavior.

It is difficult to perform operant conditioning with a range of temporal gaps between CS and US because many different behaviours can happen during the gap. We think these experiments are beyond the scope of this already large study. However, in classical conditioning it is well established that a significant learning score can result, either from a change in preference for the paired odour, or for the unpaired odour, or both (see e.g. Eschbach et al., Nature neuroscience. 2020, Extended Data Figure 2).

4) The authors should consider that given the small effects, it has to be ensured that the observed differences originate from training and are not mere pre-training biases. Though they show the pretraining results for one of the experiments (Figure 3b, the trained group), the pretraining bending is very relevant for each of the operant learning experiments. In fact, training induced effects should not only be measured by looking at the left vs right bending in the final test but as a change between pre versus post or between a trained and a mock control group. This is done for one group (Ddc-GAL4) in Figure 3b but will be mandatory for all operant learning experiments.

We have expanded Figures 4c-f and 5a-c to show bend/min towards the stimulated and unstimulated side and compute the difference between them and test whether this difference is significant (using two-sided Wilcoxon signed-rank test), both before and after training, for all genotypes and all conditions. We also compare (using a two-sided Mann-Whitney U test, with Bonferroni correction) the difference in bends/min towards the stimulated and unstimulated side (Δ bends/min), after training, between experimental and control groups (Figures 4h and 5d), as suggested by the Reviewer.

Pairing optogenetic activation of dopaminergic and serotonergic (Ddc-GAL>UAS-CsChrimson larvae, Figures 4c) or just serotonergic neurons (Tph-GAL4>UAS-CsChrimson, Figure 5a) with a specific bend direction, resulted in a significant increase in bends/min towards the stimulated side relative to the unstimulated side, after training, but not before. These results suggest that pairing a specific bend direction with the activation of serotonergic and dopaminergic neurons together, or serotonergic neurons alone, induces operant learning.

For all other genotypes there was no significant difference in bends/min to the stimulated, compared to the unstimulated side, neither before, nor after training. Thus, pairing the activation of only brain dopaminergic and sertotonergic (Ddc-GAL4>UAS-CsChrimson, teashirt-GAL80, Figure 4e), only MB dopaminergic neurons (58E02-GAL4>UAS-CsChrimson, Figures 4f), only dopaminergic neurons(TH-GAL4>CsChrimson, Figure 5b), or only brain serotonergic neurons (Tph-GAL4>UAS-CsChrimson, teashirt-GAL80, Figure 5c) with a specific bend direction did not result in a significant difference in bend direction between the stimulated and the unstimulated side, neither before, nor after training. These results indicate that none of these neuronal subsets alone were sufficient to induce operant learning.

Since all of these genotypes contained the UAS-CsChrimson transgene, these results further suggest that any putative leaky basal CsChrimson expression in some unknown neurons is not sufficient to induce operant learning.

Similarly, the yoked control larvae (Ddc-GAL4>UAS-CsChrimson, Figure 4d) in which optogenetic activation of dopaminergic and serotonergic neurons was not paired with a specific bend direction did not result in a significant difference in bend direction between the stimulated and the unstimulated side, neither before, nor after training. This shows that simply optogenetic activation of dopaminergic and serotonergic neurons unpaired with bend does not bias bend direction.

We also compared Δ bends/min after training between the experimental and control genotypes and conditions and found that they were significantly different.

Thus, after training, there was a significant difference in Δ bend/min between

– the Ddc-GAL4>UAS-CsChrimson and the Ddc-GAL4>UAS-CsChrimson, teashirt-GAL80 (Figure 4h).

– the Ddc-GAL4>UAS-CsChrimson and the 58E02-GAL4>UAS-CsChrimson (Figure 4h).

– the Ddc-GAL4>UAS-CsChrimson paired larvae and the yoked controls (Figure 4h).

– the Tph-GAL4>UAS-CsChrimson and the Tph-GAL4>UAS-CsChrimson, teashirt-GAL80 (Figure 5d).

These results are consistent with the idea that only the complete set of serotonergic neurons is sufficient to induce operant learning. In the absence of activation of, either all serotonergic neurons, or just nerve cord serotonergic neurons, we did not observe operant learning.

Finally, to demonstrate the reproducibility of operant learning, we also repeated the learning experiments on a different, low-throughput, single-larva tracker (Schulze et al., 2015) by pairing a specific bend direction with red light, either in the Ddc-GAL4>UAS-CsChrimson larvae, or in larvae in which UAS-CsChrimson was present but Ddc-GAL4 was absent (Figure 4–figure supplement 3).

As before, we found that pairing optogenetic activation of dopaminergic and serotonergic neurons (Ddc-GAL>UAS-CsChrimson larvae) with a specific bend direction, resulted in a significant increase in bends/min towards the stimulated side relative to the unstimulated side, after training, but not before, indicating operant learning has occurred (Figure 4–figure supplement 3d).

In contrast, there was no significant increase in bends/min towards the stimulated side relative to the unstimulated side, neither after, nor before training in the UAS-CsChrimson effector control larvae (Figure 4–figure supplement 3e). These results further support the idea that any putative leaky basal CsChrimson expression in some unknown neurons is not sufficient to induce operant learning.

Furthermore, after training, there was a significant difference in Δ bend/min between Ddc-GAL4>UAS-CsChrimson and the UAS-CsChrimson control larvae (Figure 4–figure supplement 3f). This confirms that optogenetic activation of Ddc serotonergic and dopaminergic neurons can induce operant learning.

These results and quantifications are shown in the new Figure 4–figure supplement 3.

5) It would improve the accessibility of the learning induced change of behavior if the authors could show the pre vs post training results for each run (10-12 larva in a plate). Further, they should plot the numbers of reinforced behaviors in each of the training protocols and relate it to the test performance.

We performed a per-run analysis on the multi-larva Ddc data. Average sample size, per run was 5 animals, so we can’t run statistics on this basis. To address the suggestion of plotting the number of reinforced behaviours in each training protocol and relate it to test performance, we have included the new Figure 4 – Supplement 2 that shows the full experiment data for each genotype in Figure 4. This full data includes all training windows in between the tests.

6) The presented data clearly suggests a decrease of the unstimulated bending rather than a change in the reinforced behavior. Though the authors mention it, they do not explain or discuss it. It will be very important for the logic of the manuscript that the authors explain this phenomenon and how it relates to operant conditioning (see point 3).

In the last paragraph of the Results section titled “Fictive reward can facilitate operant conditioning of larval bend direction”, we further discuss the observed result in the context of existing evidence for absence learning in larval classical conditioning.

“Further dissection of bend rates to each side showed that, after training, bend rates averaged together for larvae that received uncorrelated training (mean=4.80, sd=2.39, n=160) were indistinguishable from the rate of pair-trained larvae bending to the previously stimulated side (mean=4.84, sd=2.40, n=143) (p>.5; Figure 4G). However, larvae that received uncorrelated training showed a significantly higher bend rate (mean=4.80, sd=2.39, n=160) compared to pair-trained larvae bending to the previously unstimulated side (mean=4.10, sd=2.23, n=143) (p=.0037, CLES=.58; Figure 4G). This suggests that the pair-trained Ddc>CsChrimson larvae have learnt to avoid the unstimulated side. There is growing evidence from classical conditioning that larvae can learn that an unpaired CSpredicts the absence of reinforcement (Schleyer et al., 2018; Eschbach et al., 2020). Perhaps larvae are also forming memories of opposite valence in our operant conditioning paradigm, bending less to the unstimulated side because bending to that predicts the absence of appetitive Ddc activation.”

7) Though the manuscript discusses most of the data carefully, in my view the authors miss an important issue: it remains to be shown if fly larva are capable of operant learning using external reward or punishment. The presented evidence is based on artificial activation of neurons, which arguably is a hint but not a prove that operant conditioning is withing the repertoire of a fly larva, an issue the authors should mention and discuss.

We have added new sentences in the Discussion stating the limitations of artificial stimuli:

“It will also be important to determine whether conditioning with a fictive odour stimulus changes larval responses to natural odours. For example, larvae showing reduced attraction to Or42b activation may show reduced attraction to natural odours that activate Or42b neurons (e. g. ethyl acetate)”

“There are limitations to interpreting behaviour following fictive neuronal activation. It will therefore be important to replicate the new paradigms developed with this rig using lower-throughput methods with natural stimuli. For example, one might modify our multi-larva tracker to accommodate closed-loop presentation of real, innately appetitive gustatory stimuli by using closed-loop control of microfluidic devices”

Reviewer #2 (Recommendations for the authors):In addition to the points raised in the public review:– In figure 2, I was surprised to see that the responses elicited by the same Gal4 are so different when stimulated optogenetically or thermogenetically for both lines. Can the authors try to give an explanation?

The finding that different activators have different effects on the levels and dynamics of behaviour evoked could, in part, be explained by the different biophysical properties of these channels. The amount of current that enters, saturation kinetics and the duration of channel opening are all likely different. This could influence the probability and duration of each action.

By fitting a heating camera onto our system, we also show that thermogenetic activation is slower than optogenetic and different rates of activation could also influence the behavioural response.

Additionally, the sensory modality used to activate the neurons (light or mild heat) could interact with the pathways that are being activated linearly or non-linearly, resulting in distinct probability and dynamics of behaviour. For example, the increased bending in control animals (attp2>CsChrimson), in response to IR stimulation compared to red light stimulation, suggests that mild heat is more aversive.

We have added a paragraph in the Results section entitled “Proof-of-principle experiments verify multi-larva training rig's stimulation efficiency” discussing some of these possibilities:

“The quantitative difference in these rolling responses compared to optogenetic activation of the same Gal4 drivers (Figure 2D) is not surprising. These effects are likely mediated by differing biophysical properties of CsChrimson and dTrpA1 channels including single channel conductance and open state lifetime (Pulver et al., 2009, Vierock et al., 2017). In further contrast to the proof-of-principle optogenetic experiments, the slower kinetics of tissue heating caused a ca. 4-5s second temporal delay between stimulus onset and behaviour onset (Figure 2E).”

And

“The high bending frequency of attP2>dTrpA1 control larvae during IR-induced stimulation is likely indicative of mild heat aversion.”

– I am intrigued by what would be the US under natural conditions. In the discussion the authors say that they did not manage to condition the larvae using sugar sensing neurons, could it be that in *Drosophila* larvae operant conditioning only works for sensory signals imputing directly to the VNC? Have they tried conditioning the larvae by associating one side bending with optogenetic activation of noxious stimuli (without eliciting rolling)? It might be good to expand in the discussion on the expected sensory nature of US enabling operant conditioning.

This is a very interesting idea, but the identity of teste-sensing neurons that may project to the VNC (maybe a subset of external sensory neurons) is not known. In order to look for sensory modalities that mediate reward signals in both operant and classical conditioning in *Drosophila* larvae, further systematic functional characterisation of individual peripheral sensory neurons is required, as well as the development of appropriate GAL4 lines for specific subsets.

We have elaborated on this point in the Discussion section entitled “Automated operant conditioning of *Drosophila* larvae” and suggested mechanisms by which it can be explored in the future:

“It is noteworthy that Nuwal et al., 2012 used optogenetic activation of sugar-sensing neurons to establish a walking direction preference in adult *Drosophila*. Although we were unsuccessful in using a similar US to condition larval bend direction (data not shown), our Ddc and Tph results suggest that other sensory rewards may mediate operant conditioning if the information retrieval occurs within the VNC itself. Investigating *FoxP* and *pkc* expression in larval neurons may aid in identifying candidate neuronal populations, given both genes' involvement in operant self-learning in adult *Drosophila* (Mendoza et al., 2014, Brembs and Plnendl 2008, Colomb and Brembs 2016).”

Finally, we had also performed preliminary experiments in which nociceptive MD class IV neurons targeted by the ppk-Gal4 driver were optogenetically activated during bends to one side to see whether these neurons could evoke aversive operant learning. We were unable to elicit an operant learning effect in these pilot experiments, as evidenced by no significant difference in bend rate between the previously stimulated and unstimulated sides after training (preliminary data not shown ). We cannot exclude that trying a range of different stimulation and training conditions with MD class IV neurons could give a positive result in the future, but it was beyond the scope of this study to explore this.

Reviewer #3 (Recommendations for the authors):Recommended Revisions:The experiments are solid and rigorously done. Most of the issues can be resolved with text clarification or additional discussion.The best interpretation of the behaviors induced by expression in many dopaminergic and serotoninergic neurons is unclear: different neurons can have local functions – sometimes conflicting ones – so I am surprised that a very broad driver like DDC doesn't set up conflicting or non-specific reinforcement. I am also amazed that the timing works – a reward right after the bend could generate a different association than a potential punishment right before it. The interval between bends and the timing of the optogenetic reinforcement (assumed to be positive) relative to those bends should be discussed.

Within the Materials and methods section under “High-throughput closed-loop tracker > Optogenetic stimulation”, we had written “Because the LabVIEW application updated DMD projections at 20 Hz, the delay between behaviour detection and closed-loop optogenetic stimulation of individual larvae did not exceed 50 ms*.*” There was no additional delay added into the protocol. We have now added an additional sentence, below, to clarify this point within the Results section entitled “*Operant conditioning of larval bend direction*”:

“The time between the tracker detecting a left bend and light onset was no longer than 50ms.”

How many behavioral actions do you miss applying reward or punishment to? There are multiple larvae on the plate, and the bends must be detected and then the larva targeted – when you look at the videos later, what fraction of the bends that occur during the training periods were successfully reinforced?

As part of system development, we manually validated behaviour classifier performance. Within Table 1 (see Materials and methods), we show recall and precision values for several behaviours including bending. Bending recall with our software is 96.4%. We can therefore expect that 96.4% of all bends were detected and 3.6% were false negatives. The precision is 95.6%, i.e. 4.4% of detected bends were false positives. When correct bends were detected (i.e. true positives), the accuracy of left vs. right bending detection was 97.3%. In other words, only 2.7% of correctly identified bends were classified with the incorrect bending direction.

How was the training regimen selected? What about a longer, continuous period – or shorter breaks? Was there a minimum number of reinforced bends that had to occur to include a larva in the subsequent evaluation? Was the possibility that the un-reinforced epochs in the training regime result in forgetting (or learning that action does not reliably produce reward) considered?

This is a very interesting point and worth exploring in the future. We tried very few training regimens and since we obtained promising results, we continued to use the same protocol. We chose 4x3 min training rounds as we expected to get ca. 20 reinforced bends in that time period. There is a lot of scope for trying many different training regimens in the future. In particular, increasing the number of reinforced bends could yield better learning.

We do agree larvae could be extinguishing the memory in between training rounds. We introduced these breaks because we wanted to re-center the larvae that crawled to the edges. If some other method were used to prevent larvae from crawling to the edge, then these breaks could be omitted.

In this study, we did not select larvae for analysis based on a minimum number of reinforced bends during training, although this could be a good idea for future studies.

Important missing control: There have been reports that the UAS-Chrimson can have some basal/leak expression. What does the attP2>Chrimson + retinal control look like in the biasing for bend direction experiments (Figure 3B)? Even a low level of expression could serve as a reward and so this control should be tested. (The uncorrelated light control is also good, but does not rule out contribution from activation of non-serotonergic neurons.)

To address this point and the point about the need to reproduce our results in a different way, we reproduced the operant learning experiment using optogenetic activation on a different, low-throughput, single-animal tracker (Schulze et al., 2015). Using this system we paired specific bend direction with red light, either in *Ddc-GAL4, UAS-CsChrimson* larvae, or in the effector control larvae that had the *UAS-CsChrimson* transgene but not the *Ddc-GAL4* (Figure 4—figure supplement 3).

As before, we found that pairing optogenetic activation of dopaminergic and serotonergic neurons (Ddc-GAL>UAS-CsChrimson larvae) with a specific bend direction resulted in a significant increase in bends/min towards the stimulated side relative to the unstimulated side, after training, but not before, indicating operant learning has occurred (Figure 4–figure supplement 3d).

In contrast, there was no significant increase in bends/min towards the stimulated side relative to the unstimulated side, neither after, nor before training in the UAS-CsChrimson effector control larvae (Figure 4–figure supplement 3e). These results support the idea that any putative leaky CsChrimson expression in some unknown neurons is not sufficient to induce operant learning.

Furthermore, after training, there was a significant difference in Δ bend/min between Ddc-GAL4>UAS-CsChrimson and the UAS-CsChrimson larvae (Figure 4–figure supplement 3f). This confirms our finding that optogenetic activation of Ddc serotonergic and dopaminergic neurons can induce operant learning.

We have added new Figure 4 – figure supplement 3 with these results and a new section in the Results:

“We also used a previously developed, low-throughput, single-larva, closed-loop tracking system to test the reproducibility of this result on a different system (see Materials and methods for more details, Schulze et al., 2015). Fictive Ddc activation with this system also yielded a significant bend direction preference to the previously stimulated (mean=5.77, sd=2.71, n=109) versus previously unstimulated (mean=4.73, sd=2.73, n=109) side (p=.0043), after training. These results contrast those of control larvae that had the UAS-CsChrimson transgene but not the Ddc-GAL4. The effector control larvae show no bend prefference to either side after training. Based on these control larvae, we concluded that potential basal expression of CsChrimson in neurons outside of the Ddc expression pattern is not causing operant learning (Figure 4 —figure supplement 3).”

Finally, we also note that, only pairing of optogenetic activation of dopaminergic and serotonergic (Ddc-GAL>UAS-CsChrimson larvae, Figure 4c and Figure 4 —figure supplement 3d) or just serotonergic neurons (Tph-GAL4>UAS-CsChrimson, Figure 5a) with a specific bend direction, resulted in a significant increase in bends/min towards the stimulated side relative to the unstimulated side, after training, but not before.

For all other genotypes there was no significant difference in bends/min to the stimulated, compared to the unstimulated side, neither before, nor after training. Thus, pairing the activation of only brain dopaminergic and sertotonergic (Ddc-GAL4>UAS-CsChrimson, teashirt-GAL80, Figure 4e), only MB dopaminergic neurons (58E02-GAL4>UAS-CsChrimson, Figures 4f), only dopaminergic neurons (TH-GAL4>CsChrimson, Figure 5b), or only brain serotonergic neurons (Tph-GAL4>UAS-CsChrimson, teashirt-GAL80, Figure 5c) with a specific bend direction did not result in a significant difference in bend direction between the stimulated and the unstimulated side, neither before, nor after training. These results indicate that none of these neuronal subsets alone were sufficient to induce operant learning. Since all of these genotypes contained the UAS-CsChrimson transgene, these results further suggest that any putative leaky CsChrimson expression in some unknown neurons is not sufficient to induce operant learning.

We also compared Δ bends/min after training between the experimental and control genotypes conditions and found they were significantly different.

Thus, after training, there was a significant difference in Δ bend/min between

– the Ddc-GAL4>UAS-CsChrimson and the Ddc-GAL4>UAS-CsChrimson, teashirt-GAL80 (Figure 4h).

– the Ddc-GAL4>UAS-CsChrimson and the 58E02-GAL4>UAS-CsChrimson (Figure 4h).

– the Ddc-GAL4>UAS-CsChrimson paired larvae and the yoked controls (Figure 4h).

– the Tph-GAL4>UAS-CsChrimson and the Tph-GAL4>UAS-CsChrimson, teashirt-GAL80 (Figure 5d).

These results are consistent with the idea that only the complete set of serotonergic neurons is sufficient to induce operant learning. In the absence of activation of either all serotonergic neurons, or just nerve cord serotonergic neurons, we did not observe operant learning.

line 289 (and line 395): The use of necessity and sufficiency is a little confusing here. These are all ectopic manipulations. Optogenetic activation of all dopaminergic and serotoneric neurons (expressing DDC-GAL4, UAS-Chrimson) is capable of inducing increased bending. Activation of dopaminergic and serotonergic neurons in the brain alone (DDC-GAL4, UAS-Chrimson, tsh-LexA, LexOp-GAL80) does not. We don't know much about how these neurons contribute to normal operant conditioning, just that the ones in the VNC seem to be the critical ones for the optogenetic effect. It could be said that activation of serotonergic VNC neurons is sufficient to induce the optogenetic effect – but I would avoid any speculation of actual necessity. (line 375 described operant conditioning as "impaired" which is also confusing. The artificial operant conditions failed when VNC serotonergic neurons were not included. That is not the same thing.) line 413: optogenetic activation CAN serve, not DOES serve…. This all may be a difference of word use conventions among research studying learned vs. innate behaviors, but given the discussion about value of these terms (Gomez-Marin 2017, Yoshihara and Yoshihara 2018), it might be good to maximize clarity here.

We agree with these suggestions and have rephrased the relevant sections of the paper accordingly.

In response to the comment, we no longer state that our results “highlighted the necessity of dopaminergic or serotonergic neurons in the VNC for the formation of a bend direction preference”.

Instead, we now say that “dopaminergic and serotonergic neurons in the VNC appear critical to the bend direction preference formed following paired optogenetic activation of all Ddc neurons”. This new text can be found on line 425-426 of the new submission.

We have also changed the focus of our discussion of serotonergic neurons in the VNC and operant conditioning. We no longer state that “activation of the brain and SEZ is sufficient for classical conditioning, whereas the VNC is necessary for operant conditioning”.

Instead, we say that “Future experiments exclusively targeting a single serotonerig neuron per VNC hemisegment could be valuable in determining whether they are sufficient for operant learning.” This new text can be found on line 484-485 of the new submission.

Operant conditioning is measured as an increase in number of bends per minute or in the probability of bending toward the rewarded side. These are not directly comparable metrics to the performance index used in classical conditioning, but the operant effects, while statistically significant, seem very small (eg. Figure 4b vs. d). Is this an accurate observation and is there a useful contrast proposed?

We do agree that operant learning seems weaker in this case than classical conditioning. This could be a true limitation of the operant learning circuits compared to classical conditioning ones in the larva. Alternatively, it could be due the vast accumulated experience in the field about the best stimulation protocols and conditions for classical conditioning. It is likely that by exploring many different operant protocols in the future, far stronger learning could be induced.

I found the shifts between dopaminergic and serotonergic neurons, and between classical and operant conditioning experiments, a little challenging to follow. Please consider clarifying how the classical conditioning experiments contribute to the focus of this research that is suggested by the title – serotonergic neurons capable of operant conditioning. (Since it is clear early on that operant and classical conditioning are achieved by different neurons, it is not logical that the classical conditioning screen for subsets of dopaminergic or serotonergic neurons would turn up any useful candidates for dissecting operant circuits.)

We agree with this point. To address it, we have removed the manual classical conditioning screen experiments from the paper to reduce confusion. Instead, we have replaced them with a demonstration of a novel form of classical conditioning using our automated tracking system. This also addresses the flaw in the prior version that no real use was demonstrated for the combination of thermogenetic and optogenetic stimulation modules in learning.

We have adjusted the focus of the manuscript to highlight the advantages of our novel high-throughput multi-larva training system in demonstrating two new subtypes of associative learning in larvae: trace conditioning and operant learning.

We have added the new Figure 3 and the new section entitled: “Aversion to fictive Or42b develops after forward-paired trace conditioning”.

“Having verified the efficacy of optogenetic and thermogenetic stimulation in our system, we first studied whether these methods could be used to train larvae in a previously unexplored classical conditioning task that requires precise temporal control of both CS and US. […] These results show that our tracker can be used to perform automated, high-throughput classical conditioning in *Drosophila* larvae. To the best of our knowledge, these results also provide the first evidence that larvae can perform classical conditioning with significant (9s) offset-to-onset gaps between stimuli (i.e. trace conditioning).”

The Tsh-LexA, lexOp-GAL80 combination should reference Simpson 2016 J Neurogenetics; only the Tsh-GAL80 was published in Clyne and Miesenbock.

The Simpson 2016 J Neurogenetics reference has now been included in the text. Specifically, we now say that “[we] took an intersectional approach by targeting these transgenes with the LexA/LexAop binary system (Simpson, 2016) and expressing CsChrimson in Ddc neurons using Gal4/UAS”. This text can be found on line 409 of the new submission..